# Light regulates nuclear detainment of intron-retained transcripts through COP1-spliceosome to modulate photomorphogenesis

Hua Zhou [1], Haiyue Zeng [2,3], Tingting Yan[1], Sunlu Chen [4], Ying Fu[2], Guochen Qin [2], Xianhai Zhao [1], Yueqin Heng[1], Jian Li [1], Fang Lin [5], Dongqing Xu [4], Ning Wei [6] & Xing Wang Deng [1,2,3] ✉

Intron retention (IR) is the most common alternative splicing event in *Arabidopsis*. An increasing number of studies have demonstrated the major role of IR in gene expression regulation. The impacts of IR on plant growth and development and response to environments remain underexplored. Here, we found that IR functions directly in gene expression regulation on a genome-wide scale through the detainment of intron-retained transcripts (IRTs) in the nucleus. Nuclear-retained IRTs can be kept away from translation through this mechanism. COP1-dependent light modulation of the IRTs of light signaling genes, such as *PIF4*, *RVE1*, and *ABA3*, contribute to seedling morphological development in response to changing light conditions. Furthermore, light-induced IR changes are under the control of the spliceosome, and in part through COP1-dependent ubiquitination and degradation of DCS1, a plant-specific spliceosomal component. Our data suggest that light regulates the activity of the spliceosome and the consequent IRT nucleus detainment to modulate photomorphogenesis through COP1.

In eukaryotic organisms, alternative splicing (AS) is the process of differential intron removal and exon ligation in a single precursor mRNA (pre-mRNA) to generate multiple mRNA products[1]. AS can be divided into four main types, namely, alternative 3' splice sites (A3SS), alternative 5' splice sites (A5SS), exon skipping (ES), and intron retention (IR)[2]. AS functions directly in generating transcriptomic diversity and complexity[1,3]. In *Arabidopsis*, 42–61% of intron-containing genes are subjected to AS, thus suggesting the essential role of AS in regulating plant growth and development[4–6]. IR is the most widespread AS event in plants, and it functions directly in

gene expression regulation[2]. To date, a widely accepted role of IR in plant gene regulation is to give rise to nonsense-mediated decay (NMD), triggered by premature termination codons (PTCs)[7]. However, recent studies have discovered that intron-retained transcripts (IRTs) are mostly detained in the nucleus and therefore avoid NMD[5,8–11]. Upon external stimulus and developmental phase changes, the unspliced introns in IRTs can be removed post-transcriptionally in a spliceosome-dependent and transcription-independent manner[10,11]. After that, the fully spliced transcript isoforms translocate into the cytoplasm and are competent for translation[11]. In this

[1]Shenzhen Key Laboratory of Plant Genetic Engineering and Molecular Design, Southern University of Science and Technology, Shenzhen 518055, China. [2]Peking University Institute of Advanced Agricultural Sciences, Shandong Laboratory of Advanced Agricultural Sciences at Weifang, Weifang 61000 Shandong, China. [3]Peking-Tsinghua Center for Life Sciences, Peking University, 100871 Beijing, China. [4]State Key Laboratory of Crop Genetics and Germplasm Enhancement, Nanjing Agricultural University, Nanjing 210095, China. [5]Ministry of Education Key Laboratory of Cell Activities and Stress Adaptations, Lanzhou University, Lanzhou 730000, China. [6]School of Life Sciences, Southwest University, Chongqing 400715, China. ✉e-mail: deng@pku.edu.cn

mechanism, IR downregulates gene expression through the storage of IRTs in the nucleus[11–13]. However, the extent to which this mechanism contributes to gene expression regulation and plant developmental control is unclear. AS is accomplished by the RNA–protein complex called the spliceosome[13–15]. The spliceosome, which accurately assembles at pre-mRNA, is a megadalton complex consisting of five small nuclear RNAs (snRNAs) and over 200 associated proteins[5,15–18]. Both the conformation and composition of the spliceosome are highly dynamic and under precise control in all organisms to ensure accuracy and flexibility during splicing[19]. In humans, around one-third of all known disease-causing mutations disrupt normal pre-mRNA splicing[20,21]. Dysfunctional mRNA splicing is widespread in cancer, and dysregulated expression or mutations in splicing factors are known to contribute to tumorigenesis[13,22]. In plants, a deficiency in splicing factors can lead to developmental defects or even embryonic lethality[23–30]. AS is involved in many interactions between plants and environmental factors, especially in responding to changing light conditions. Accumulating data have shown that light modulates AS in plants in various ways[31,32]. First, as an energy source, light can induce AS through the glucose–TOR pathway[33]. Photosynthesized sugars function as mobile signals to coordinate AS responses to light throughout the whole plant[33]. Second, light can regulate AS via a chloroplast retrograde signaling pathway in connection with the PQ pool[34]. It has been reported that light can trigger AS in a phytochrome-dependent manner[3]. During de-etiolation, 6.9% of the annotated genes in *Arabidopsis* undergo rapid changes in their AS patterns[3]. The AS of light signaling genes is important for photomorphogenic development[3,35]. Overexpression of the phytochrome-induced isoforms of *SAP3* promotes photomorphogenesis in *Arabidopsis*[3]. phyB-induced *PIF3* AS-uORF promotes light responses by inhibiting *PIF3* mRNA translation[35]. Light-responsive IRTs can form heterodimers with their annotated full-length isoforms to regulate seedling photomorphogenesis[36]. Accordingly, dysfunction of splicing factors, such as RRC1, SFPS, and SWAP1, triggers impaired phyB-dependent photomorphogenic responses[26,28,30]. Overall, these data demonstrate that photomorphogenic responses in plants can be controlled by gene regulation at the mRNA splicing step performed by the spliceosome. However, the extent to which AS regulation contributes to seedling photomorphogenesis and its response to environmental changes and stress treatments remains largely unknown. COP1, an E3 ubiquitin ligase, acts as a tightly controlled switch that determines the conversion from etiolation to de-etiolation in *Arabidopsis*[37]. In the dark, COP1 ubiquitinates numerous positive regulators of the light signaling pathway, such as HY5 and its homolog HYH, for degradation to repress light-triggered development[37]. Upon light illumination, photoreceptors, including cryptochromes and phytochromes, directly interact with COP1 to negatively regulate its activity[37–40]. In this study, through mapping and whole-genome re-sequencing of 13 dominant *cop1-6* suppressors identified from EMS mutagenesis, we cloned 6 *Dominant cop1-6 Suppressor* (*DCS*) genes. Phylogenetic analyses and IP/MS data indicated that all the DCS proteins were components of the *Arabidopsis* spliceosome. Furthermore, genome-wide mRNA-seq analysis demonstrated that COP1 was involved in light-regulated AS, mainly IR, to coordinate seedling photomorphogenic development. IR regulates gene expression through detainment of their related IRTs in the nucleus, preventing their translation. Light-dependent regulation of light signaling genes, such as *PIF4*, *RVE1*, and *ABA3*, in IRTs contributes to seedling photomorphogenic development through COP1. Further, COP1 adjusts the functional state of the spliceosome by directly interacting with and ubiquitinating DCS1, a plant-specific spliceosome component. Thus, we reason that light modulates the activity of the spliceosome through COP1 to regulate seedling photomorphogenic development.

## Results

### Identification and map-based cloning of Dominant cop1-6 Suppressor (DCS) genes

COP1, a negative regulator of light signaling, is essential for plant growth and development[37]. While *cop1* null mutants are seedling lethal, the *cop1-6* allele mutant is viable, which produces a small quantity of a mutated COP1 protein with five additional amino acids inserted in frame between Glu-301 and Phe-302[41]. *cop1-6* plants display a constitutive photomorphogenic phenotype with short hypocotyls and expanded cotyledons when grown in the dark[41]. To better understand the basis of COP1 function, we performed a genetic screen to identify the suppressors of *cop1-6* in the dark. In addition to the published recessive *cop1-6* suppressor mutants[42–47], we identified and isolated 13 additional extragenic dominant suppressors of *cop1-6* (Fig. 1a). When grown in the dark, the hypocotyl lengths of all the dominant suppressor mutants were drastically longer than those of *cop1-6* (Fig. 1a). However, the suppression degrees showed some differences among these dominant suppressor mutants, especially in the degree of cotyledon opening of *dcs2 cop1-6* mutants (Fig. 1a). In *dcs2-1 cop1-6* and *dcs2-3 cop1-6*, Gly367 and Gly495 were mutated to Glu and Arg, respectively. Both Glu and Arg were charged amino acids. In *dcs2-2 cop1-6* and *dcs2-4 cop1-6*, Ala457 and Asp512 were mutated to uncharged amino acids (Fig. 1b). There may be the differences in the physical properties of amino acids caused the different degrees of suppression.

Next, we carried out map-based cloning combined with whole-genome re-sequencing strategies for the 13 dominant mutations (Supplementary Fig. 1). Together, we identified six genes that we named *Dominant cop1-6 Suppressor 1* to *6* (*DCS1–DCS6*) (Fig. 1b). All of the dominant mutants carried a missense mutation in their protein sequences, except for a nonsense mutation in *dcs1-3* (Fig. 1b, c).

We performed genomic complementation experiments to verify that the point mutations in DCS proteins indeed cause the *cop1-6* suppression phenotype. The mutated *DCS* genes were transformed into the *cop1-6* mutant background. All of the complemented seedlings displayed hypocotyl lengths similar to the corresponding *dcs cop1-6* mutants when grown in the dark, with each having a drastically longer length compared with *cop1-6* (Supplementary Fig. 2). These results demonstrate that the mutations in *DCS* genes can complement the phenotype conferred by *cop1-6*.

### DCS proteins are components of the Arabidopsis spliceosome

Phylogenetic analyses of DCS proteins revealed that DCS2, DCS3, DCS5, and DCS6 were conserved across fungi, plants, and animals, while DCS1 and DCS4 were apparently plant-specific proteins (Supplementary Fig. 3a–c and Supplementary Table 1). DCS3, DCS5, and DCS6 have been reported to be spliceosomal components (Supplementary Fig. 3a)[23,24,48]. *DCS3* encodes a highly conserved and essential component of the U5 snRNP, Prp8, which occupies a central position within the catalytic core of the spliceosome (Supplementary Fig. 3a)[48,49]. *DCS5* encodes the *Arabidopsis* homolog of yeast step II splicing factor Slu7, which is involved in 3′ splice site selection (Supplementary Fig. 3a)[24,50]. *DCS6* encodes the homolog of human CACTIN, a component of the catalytically active spliceosome complex C, which performs the second step of splicing (Supplementary Fig. 3a)[23,51]. In human spliceosome, both Slu7 and CACTIN directly interact with Prp8[52].

Protein structure analysis indicated that DCS2 and DCS4 were members of DEAD-box helicases, a large family of conserved RNA-binding proteins (Fig. 1c). DEAD-box helicases have been proposed to drive the conformational rearrangements of the spliceosome as it goes from one stage to the next in the splicing cycle[53]. Human DDX41, homolog to DCS2, is specifically recruited to the catalytically active spliceosome C complex (Supplementary Fig. 3a)[54,55]. DCS1 has been identified as an RNA-binding protein[56]. Further, in our analysis, DCS1

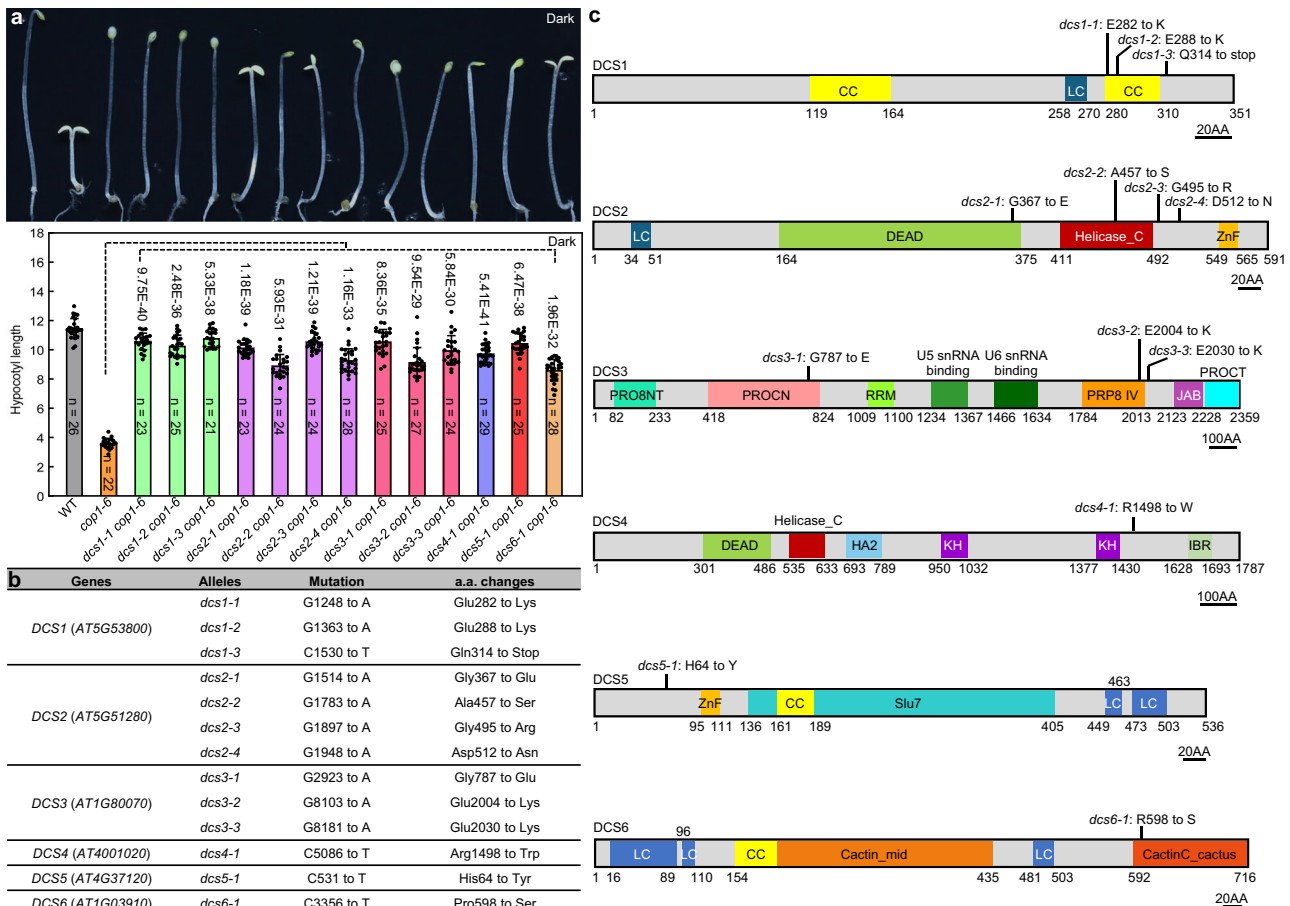

**Fig. 1 | Isolation and identification of *cop1-6* dominant suppressors. a** *dcs* mutants suppress the constitutively photomorphogenic phenotype of *cop1-6* in the dark. *Arabidopsis* seedlings were grown on MS plates in the dark for 5 days, and then the hypocotyl phenotype and length (mm) were analyzed. Values are mean ± SEM. Significance is evaluated by the two-sided Student's *t* test and *P* values are indicated above the bars. **b** Mutations identified in the *dcs* mutants and the consequences of the mutations in the DCS proteins. **c** Protein structures of DCS proteins. The positions of the point mutation sites are labeled by black lines. CC coiled coil, LC low complexity. Source data are provided as a Source Data file.

interacted with the core splicing factors SR45, U1-70K, and SKIP (Supplementary Fig. 3d).

Until now, the spliceosome machinery has not been well characterized in *Arabidopsis*. To investigate whether all the DCS proteins participate in the formation of the *Arabidopsis* spliceosome, we performed immunoprecipitation-mass spectrometry (IP-MS) analysis using *DCS* overexpression lines and studied the DCS interactomes. In brief, among the 395 putative splicing factors and related proteins[3], dozens of spliceosome members were immunoprecipitated by every DCS protein (Supplementary Table 2). Interestingly, DCS3 was immunoprecipitated by each of the tested DCS proteins (see below). These findings indicate that DCS proteins are components of the *Arabidopsis* spliceosome and DCS3 might be at the heart of the spliceosome.

### Light and COP1 control genome-wide alternative splicing in Arabidopsis

Previous studies have reported that the *cop1-6* recessive suppressors *csu1* to *csu6* have distinct functions[42–47]. Given our data suggesting that all six of the DCS proteins are components of the spliceosome, we considered COP1's potential function in mRNA splicing. A recent study reported that COP1 can regulate AS during light-regulated plant growth and development[57]. To verify that COP1 is involved in light-regulated AS and to investigate the underlying mechanisms, we performed mRNA-seq analysis of 5-day (d)-old, white light-exposed WT (WT_L) and dark-grown *cop1-6* mutant (*cop1-6*_D) seedlings; 5-d-old dark-grown WT (WT_D) seedlings were used as the control (Fig. 2a).

We detected 2623 light-responsive AS events and 2443 COP1-responsive AS events (Fig. 2b, c and Supplementary Data 1). Among these, four major types, namely, IR, exon skipping (ES), alternative 3′ splice site (A3SS), and alternative 5′ splice site (A5SS), were identified (Fig. 2b, c and Supplementary Fig. 4a). Our data showed that only 7.4% light-dependent and 6.7% COP1-dependent differentially expressed (DE) genes underwent AS, respectively (Supplementary Fig. 4b, c and Supplementary Data 1, 2). Next, we performed GO analysis of the genes that displayed light- or COP1-responsive AS. The GO term related to response to light stimulus was significantly enriched in both groups (Supplementary Fig. 4d, e). Previous studies have reported that light can induce the AS of some core circadian clock genes to regulate photomorphogenesis[3,26,28,30]. However, in our data, the GO term related to circadian clock was not enriched (Supplementary Fig. 4d, e). Additionally, GO terms related to RNA splicing, mRNA processing, and spliceosome were not enriched in our data (Supplementary Fig. 4d, e), thus confirming the function of COP1 in mRNA splicing. These results indicate that both light and COP1 are involved in the genome-wide regulation of AS, which seems to be another layer of gene regulation relative to steady-state transcript expression.

COP1 is an essential negative factor in light-regulated morphological development. Hence, we compared COP1-responsive AS events with light-responsive AS events. Almost 49.4% of the COP1-responsive AS events overlapped with the light-responsive AS events in our analysis (Fig. 2d). As IR is the most prevalent event, representing around 60% of total AS events (Fig. 2b, c), we further compared COP1-

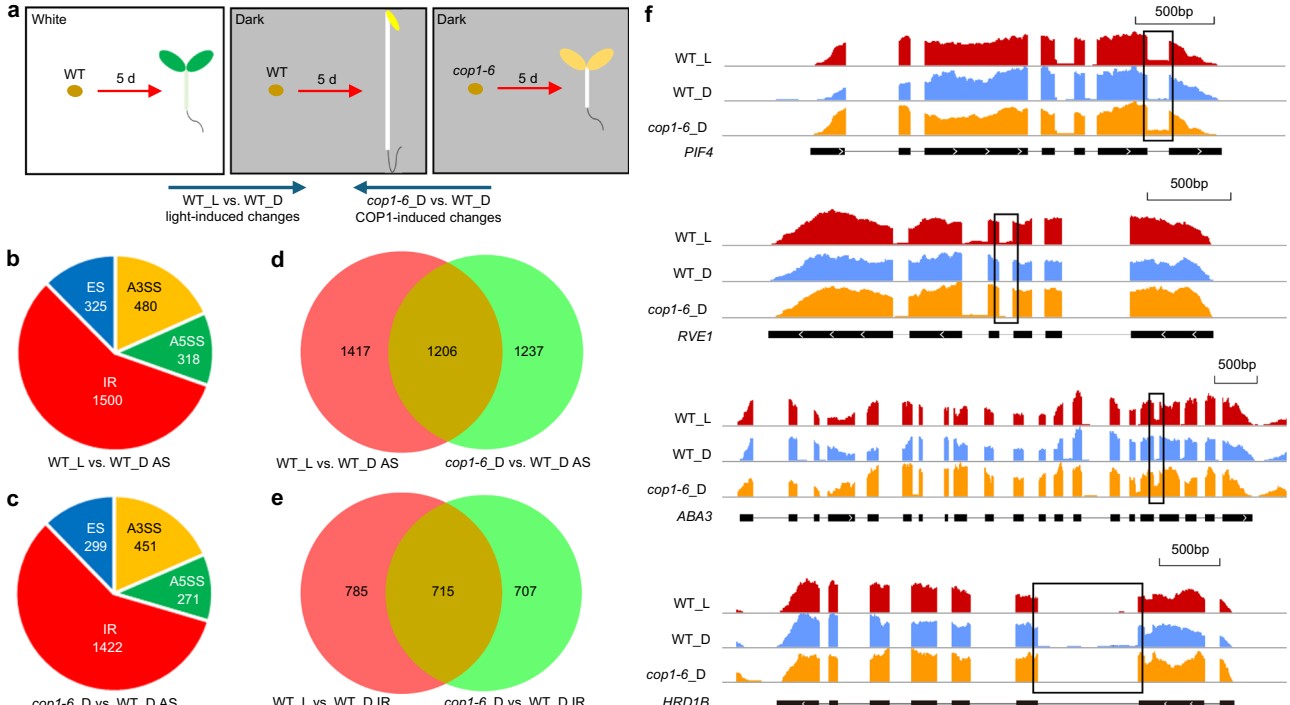

**Fig. 2 | Light and COP1 control genome-wide alternative splicing in *Arabidopsis*. a** Flowchart for the analysis of light- and COP1-induced changes in AS. For transcriptomic profiling by mRNA-seq, total RNA was extracted from 5-d-old continuous white light- (50.6 µmol m$^{-2}$ s$^{-1}$) exposed WT (WT_L) and dark-grown *cop1-6* mutant (*cop1-6*_D) seedlings, while total RNA extracted from 5-d-old WT etiolated seedlings (WT_D) was used as the control. Pie chart illustrating the number of AS events that display light- (**b**) or COP1-dependent (**c**) changes. Four major types of AS events, including IR, ES, A3SS, and A5SS, are shown. AS, *P* < 0.05. Venn diagrams illustrating the number of AS events (**d**) and IR events (**e**) that display either light-dependent changes (red), COP1-dependent changes (green), or both light- and COP1-dependent changes (yellow). AS, *P* < 0.05. **f** IR isoforms of *PIF4*, *RVE1*, *ABA3*, and *HRD1B*, identified by mRNA-seq, displaying changes in WT_L, *cop1-6*_D, and WT_D seedlings. Annotated gene structures are shown, with thick lines representing exons and thin lines representing introns. Wiggle plots showing the normalized read coverage are shown in red for WT_L, in blue for WT_D, and in yellow for *cop1-6*_D. The black frames indicate the position of the retained introns.

responsive IR events with light-responsive IR events. Approximately 50% of the IR events in both of the groups overlapped (Fig. 2e). Taken together, these data imply that COP1 participates in light-controlled AS in regulating light response, mainly through IR.

We surveyed the lists of genes with IR events responding to both light and COP1 and selected 12 genes to confirm the mRNA-seq data using semi-quantitative RT-PCR analysis. All of these genes underwent IR events, and the IRTs of these genes responded to both light and COP1 (Supplementary Figs. 5, 6a, b). Among these genes, we selected four genes, *PIF4*, *RVE1*, *ABA3*, and *HRD1B*, for further detailed analysis. *PIF4*, *RVE1*, and *ABA3* have been previously reported as negative factors in regulating seedling photomorphogenic development[58–60]. Wiggle plots of mRNA-seq data showed that IRTs of *PIF4*, *RVE1*, and *ABA3* were upregulated by light and *cop1-6*, while IRT of *HRD1B* was downregulated by light and *cop1-6* (Fig. 2f). Additionally, the splicing index (SI), defined as the abundance of the intron-retained isoform relative to the total mRNA level, was checked. We observed higher SI values for *PIF4*, *RVE1*, and *ABA3* in WT_L and *cop1-6*_D seedlings than in WT_D seedlings, while the SI value for *HRD1B* was higher in WT_D seedlings, indicating that the splicing response of these genes was regulated in the same direction by light and *cop1-6* (Supplementary Fig. 6c–f). These data also suggest that COP1 is involved in light-regulated genome-wide AS variation. Further, we found that these SI changes give rise to significant changes in the functional transcripts of *PIF4*, *RVE1*, *ABA3*, and *HRD1B* among WT_L, *cop1-6*_D and WT_D seedlings (Supplementary Fig. 7).

## Intron retention leads to detainment of intron-retained transcripts in the nucleus

Since IR is the most prevalent AS event in *Arabidopsis*, we investigated the underlying mechanism of how IR affects gene expression.

IRTs often carry premature termination codons (PTCs), which trigger nonsense-mediated decay (NMD) in the cytoplasm[9]. However, most IRTs are insensitive to the plant NMD degradation pathway[9,61]. This led us to question the cellular fate of these special transcripts, as nuclear localization would keep them away from NMD machinery.

To evaluate the subcellular localization of these IRTs, we extracted RNA from the nuclear and cytoplasmic fractions, and then performed mRNA-seq of each fraction (Supplementary Fig. 8a). In the WT_L, WT_D, and *cop1-6*_D samples, the AS variant levels for approximately 90% of the IRTs were consistently higher in the nucleus than in the cytoplasm, and among them, >59% of the IRTs were undetectable in the cytoplasmic fractions of all the three samples (Fig. 3a and Supplementary Data 1). Interestingly, we noticed that the nuclear vs. cytoplasmic data showed a big difference in the numbers of IR events in WT_L, WT_D, and *cop1-6*_D (Fig. 3a). Previous studies have reported that light enhances the global translational efficiency in *Arabidopsis*, while COP1 represses it in the dark[62,63]. This indicates that many more IRTs are fully spliced to remove the introns and then translocated into the cytoplasm for translation in WT_L and *cop1-6*_D than in WT_D. Furthermore, we checked the subcellular localization of the IRTs of *PIF4*, *RVE1*, *ABA3*, and *HRD1B* using semi-quantitative RT-PCR analysis (Supplementary Fig. 8b). The results clearly showed that the IRTs of *PIF4*, *RVE1*, *ABA3*, and *HRD1B* were almost exclusively in the nuclear fraction, whereas their intron-spliced transcripts were present in both the nuclear and cytoplasmic fractions (Supplementary Fig. 8b). To determine whether IR could lead to the nuclear detainment of its related transcripts, we investigated the subcellular localization of the IRTs of *PIF4*, *RVE1*, *ABA3*, and *HRD1B* using RNA FISH in white light-grown WT. We found that almost all of the IRTs of *PIF4*, *RVE1*, *ABA3*, and

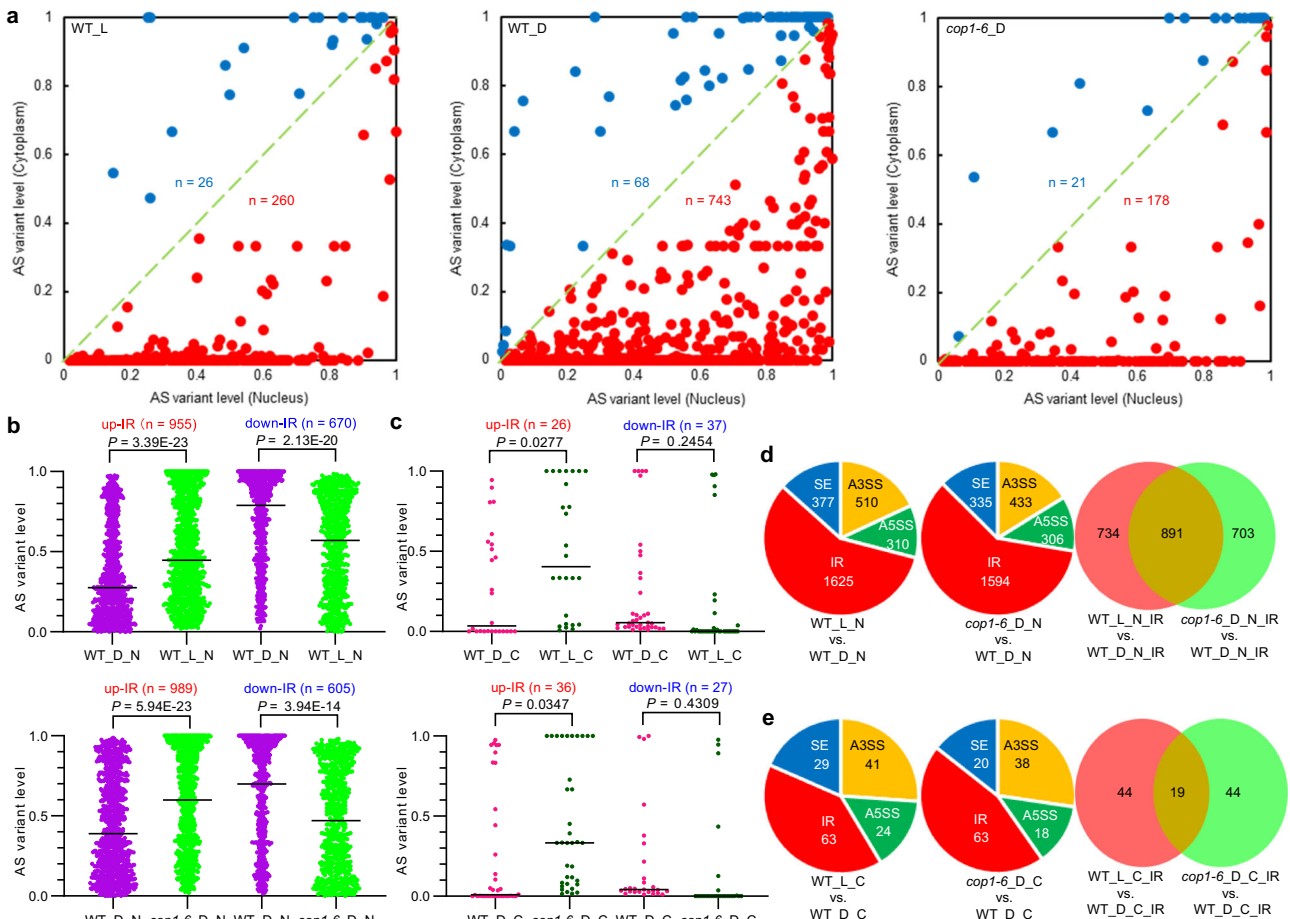

**Fig. 3 | Intron retention leads to nuclear detainment of intron-retained transcripts. a** IRTs are more abundant in the nuclear fraction than in the cytoplasmic fraction. Nuclear RNA and cytoplasmic RNA were extracted from 5-d-old WT_L (*left panel*), WT_D (*middle panel*), and *cop1-6*_D (*right panel*) seedlings. Scatter plots showing variations in IR events between the nuclear and cytoplasmic fractions of WT_L, WT_D, and *cop1-6*_D samples. AS variant level refers to IncLevel, which is calculated from normalized counts using rMATS. Histograms showing changes in light- (*top panel*) and COP1-dependent (*bottom panel*) IR events in either nuclear (**b**) or cytoplasmic (**c**) fractions. **d, e** A significant number of IR events commonly occurred in both light- and COP1-responsive seedlings. Pie chart showing the number of light- and COP1-dependent AS events in either nuclear (**d**) or cytoplasmic (**e**) fractions. Venn diagrams showing the number of IR events that display both light- and COP1-dependent changes in either nuclear (**d**) or cytoplasmic (**e**) fractions. Significance is evaluated by the two-sided Student's *t* test and *P* values are indicated. WT_L_N, nuclear RNA of WT_L; WT_L_C, cytoplasmic RNA of WT_L; WT_D_N, nuclear RNA of WT_D; WT_D_C, cytoplasmic RNA of WT_D; *cop1-6*_D_N, nuclear RNA of *cop1-6*_D; *cop1-6*_D_C, cytoplasmic RNA of *cop1-6*_D. AS, *P* < 0.05. Source data are provided as a Source Data file.

*HRD1B* located in the nucleus (Supplementary Fig. 8c). These results suggest that IR leads to the detainment of IRTs in the nucleus.

Considering the nuclear detainment of IRTs, we also analyzed nuclear IR events. A total of 1625 and 1594 nuclear IR events were responsive to light and COP1, respectively (Fig. 3b and Supplementary Data 1). Compared with the nuclear fraction, only approximately 4% of the IR events were in the cytoplasmic fraction, thus suggesting that the IR events detected in the total RNA fraction were mostly derived from the nuclear RNA fraction (Fig. 3b, c and Supplementary Data 1). Around 60% of the IRTs, including those from *PIF4*, *RVE1*, and *ABA3*, were upregulated in light-grown WT and dark-grown *cop1-6* seedlings (Fig. 3b and Supplementary Fig. 8d–f). Similarly, IR was also the most common AS event in both the nuclear and cytoplasmic fractions (Fig. 3d, e). Approximately 55% of the nuclear IR events overlapped, but only approximately 30% of cytoplasmic IR events overlapped in both the light- and COP1-responsive groups (Fig. 3d, e). Interestingly, A3SS, A5SS, and ES events were also greatly decreased in the cytoplasmic fraction compared with the nuclear fraction, probably due to the degradation of these AS transcripts by NMD in the cytoplasm[61]. These data demonstrate that COP1 is involved in light-regulated IR, which leads to the nuclear detainment of its IRTs.

**Intron retention-mediated gene expression is essential for proper morphological development to light/dark conditions**

Considering that IR downregulates gene expression through the detainment of IRTs in the nucleus, thus preventing their translation[11,13], we investigated the functions of light- and COP1-regulated IR in seedling morphological development to light/dark conditions. Light-grown WT seedlings and dark-grown *cop1-6* seedlings, both of which displayed photomorphogenic phenotypes, had higher SI values for *PIF4*, *RVE1*, and *ABA3* compared with dark-grown WT seedlings (Supplementary Fig. 8d–g). This implies that downregulation of *PIF4*, *RVE1*, and *ABA3* expression might contribute to the photomorphogenic phenotype of WT_L and *cop1-6*_D seedlings. As expected, we detected that the expression of *PIF4*, *RVE1*, and *ABA3* in the cytoplasmic fraction was lower in WT_L and *cop1-6*_D seedlings than in WT_D seedlings (Supplementary Fig. 9). Further, phenotype data showed that deficiency in PIF4, RVE1, and ABA3 at the same time can greatly reduce the seedling hypocotyl length to around 76% of the WT level when grown in the dark (Fig. 4a). However, the hypocotyl length of *pif4-2 rve1-2 aba3-3* was much longer than that of *cop1-6* (Fig. 4a). The above data showed that hundreds of IRTs were regulated by both light and COP1 (Figs. 2e and 3d). This indicates that some other light-/COP1-regulated

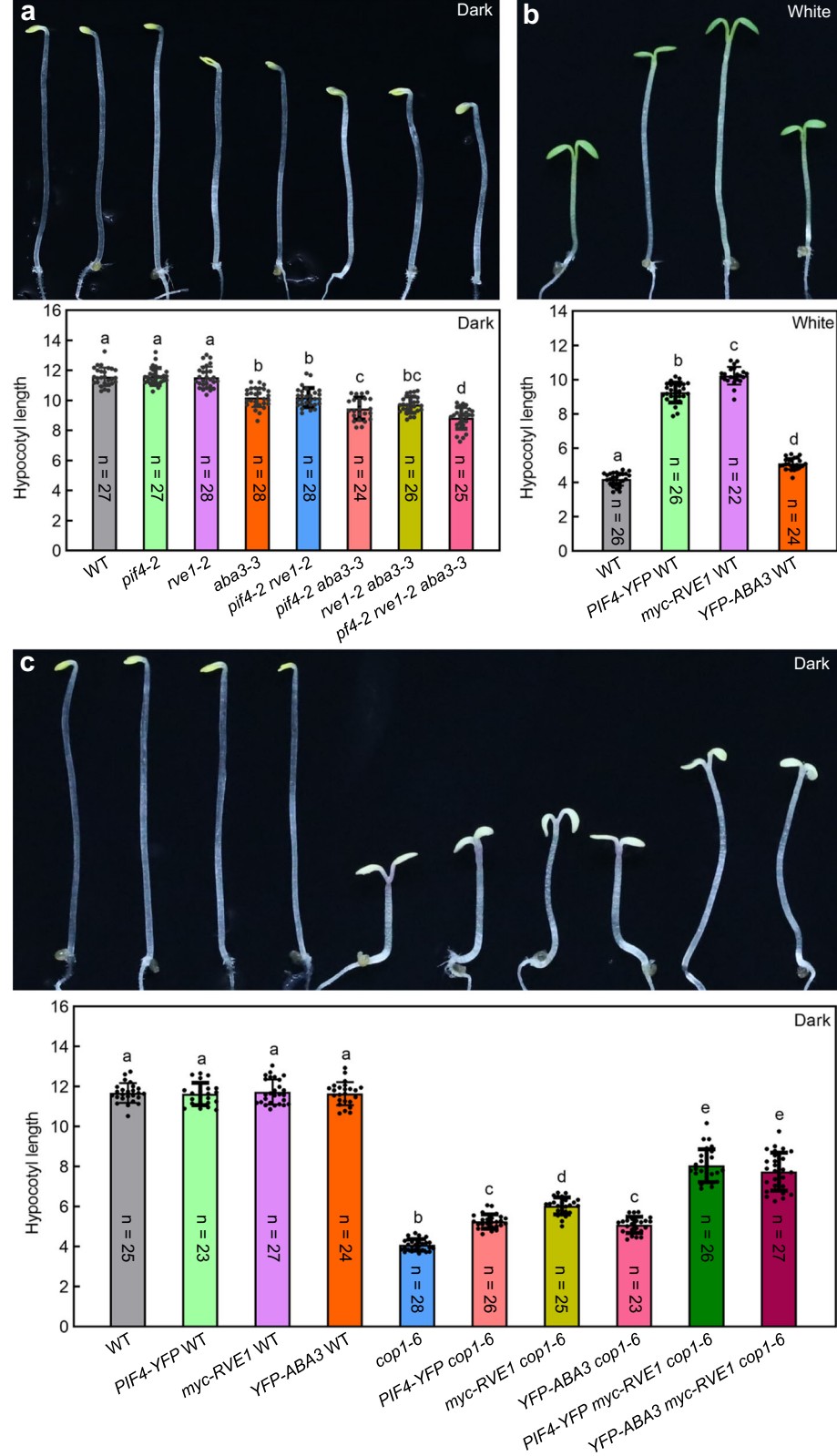

**Fig. 4 | Intron retention-mediated gene expression is essential for proper morphological development to light/dark conditions. a** Hypocotyl phenotype of single, double, and triple mutants of *pif4-2*, *rve1-2*, and *aba3-3* in the dark. Deficiencies in PIF4, RVE1, and ABA3 caused a shortened hypocotyl length in *Arabidopsis* seedlings grown in the dark. **b** Overexpression of *PIF4*, *RVE1*, and *ABA3* in the WT background promoted hypocotyl growth in continuous white light (11.2 µmol m$^{-2}$ s$^{-1}$). **c** Overexpression of *PIF4*, *RVE1*, and *ABA3* significantly suppressed the constitutively photomorphogenic phenotype of *cop1-6* when grown in the dark. *Arabidopsis* seedlings were grown in white light (**b**) or in the dark (**a, c**) for 5 days, and then the hypocotyl phenotype and length (mm) were analyzed. Values are mean ± SEM. Letters above the bars indicate significant differences (*P* < 0.05) as determined by one-way ANOVA with Tukey's post hoc analysis. Source data are provided as a Source Data file.

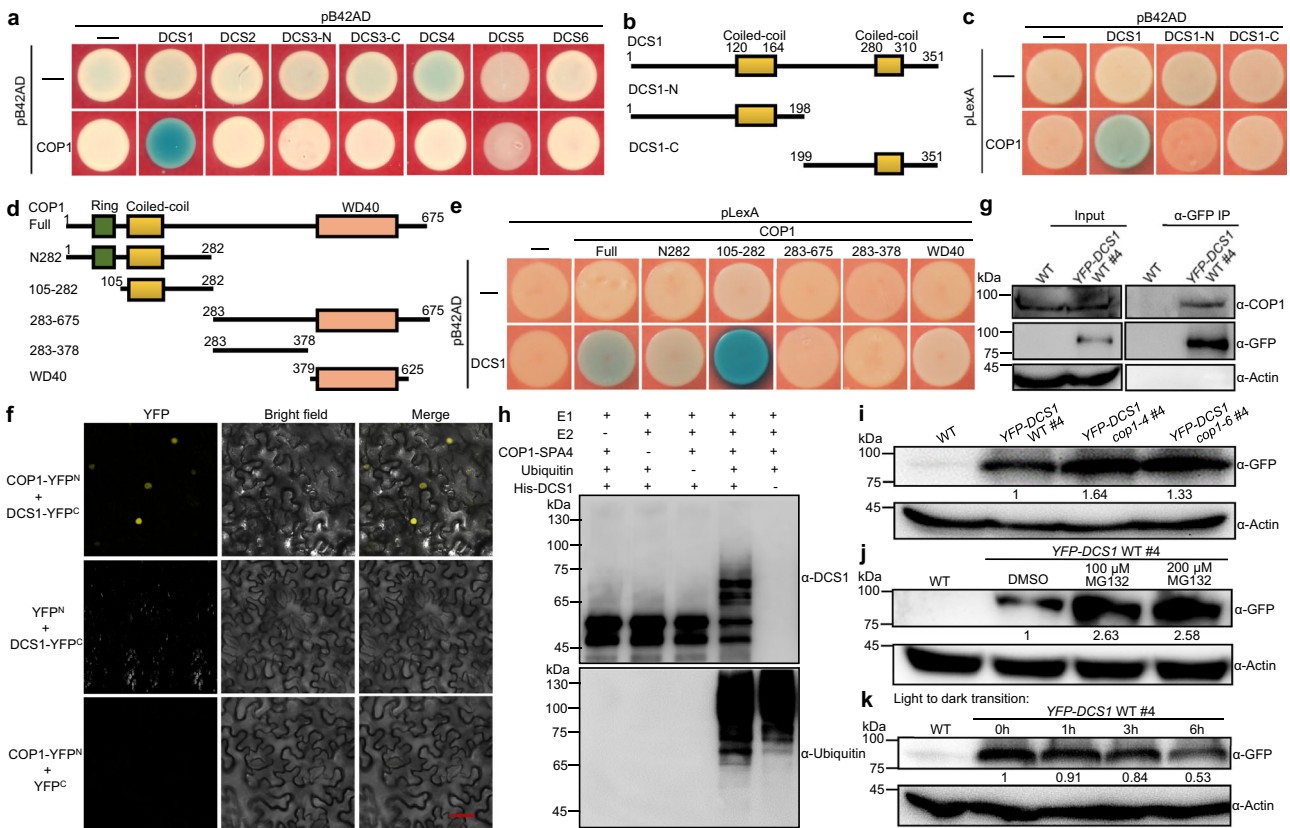

**Fig. 5 | DCS1 physically interacts with COP1 and undergoes COP1-mediated ubiquitination and degradation. a** Yeast two-hybrid screening for the interactions between DCS proteins and COP1. **b** Schematic diagram of the domain structure of DCS1 and truncated DCS1 proteins. DCS1-N and DCS1-C are the N- and C-terminal parts of DCS1, respectively. The numbers indicate the amino acid positions in DCS1. **c** Yeast two-hybrid assays showing that the whole DCS1 protein is responsible for its interaction with COP1. **d** Schematic diagram of the domain structure of COP1 and the truncated COP1 proteins (COP1-N282, COP1-105-282, COP1-283-675, COP1-283-378, and COP1-WD40). The numbers indicate the amino acid positions in COP1. **e** Yeast two-hybrid assays showing that the coiled-coil domain of COP1 is responsible for the interaction of COP1 with DCS1. **f** BiFC assays showing the interaction between COP1 and DCS1. Full-length COP1 and DCS1 were fused to split N- or C-terminal fragments of YFP (YFP$^N$ or YFP$^C$), respectively. Merge, merged images of the YFP channel and bright field. Bar, 50 μm. **g** Co-IP analysis showing that YFP-DCS1 interacts with COP1. Total proteins were extracted from the WT and YFP-DCS1 overexpressing seedlings. The immunoprecipitates were evaluated using anti-GFP and anti-COP1 antibodies. Actin served as a loading control. **h** COP1-mediated ubiquitination of DCS1. In vitro ubiquitination assays were performed using UBE1 (E1), UbcH5b (E2), COP1-SPA4 complex, and Arabidopsis ubiquitin. Ubiquitinated His-DCS1 was detected using both anti-DCS1 and anti-ubiquitin antibodies. **i** Immunoblotting analysis showing YFP-DCS1 protein levels in YFP-DCS1 WT #4, YFP-DCS1 cop1-4 #4, and YFP-DCS1 cop1-6 #4 seedlings grown in the dark for 5 d. Actin served as a loading control. **j** Immunoblotting analysis showing YFP-DCS1 protein levels in 4-d-old etiolated YFP-DCS1 WT #4 seedlings treated with various concentrations (0, 100, 200 μM) of MG132, an inhibitor of the 26 S proteasome. YFP-DCS1 overexpressing seedlings treated with DMSO served as a negative control. Actin served as a loading control. **k** Western blot analysis of YFP-DCS1 protein levels in 5-d-old white light-grown YFP-DCS1 WT #4 seedlings upon transfer to the dark at the indicated time points (0, 1, 3, 6 h). Source data are provided as a Source Data file.

IRTs, not just *PIF4*, *RVE1*, and *ABA3*, are involved in this process. Additionally, the *aba3-3*, *pif4-2 rve1-2*, *pif4-2 aba3-3*, and *rve1-2 aba3-3* mutants all exhibited shortened hypocotyl phenotypes when grown in the dark (Fig. 4a and Supplementary Fig. 10). Conversely, *Arabidopsis* seedlings overexpressing *PIF4*, *RVE1*, and *ABA3* in the WT background exhibited an elongated hypocotyl phenotype in white light (Fig. 4b). Moreover, overexpression of *PIF4*, *RVE1*, and *ABA3* in the *cop1-6* background significantly repressed the constitutive photomorphogenic phenotype of *cop1-6* when grown in the dark (Fig. 4c). Overexpression of *PIF4* and *ABA3* in the *myc-RVE1 cop1-6* background further promoted hypocotyl growth (Fig. 4c), indicating that *PIF4*, *RVE1*, and *ABA3* may have additive effects on the suppression of the *cop1-6* phenotype. Taken together, these data imply that light-/COP1-induced IR regulation of *PIF4*, *RVE1*, and *ABA3* contributes to seedling photomorphogenic development.

## The dcs suppressors partially rescue the SI of PIF4, RVE1, ABA3, and HRD1B in cop1-6

Our data indicate that the SI alterations of *PIF4*, *RVE1*, *ABA3*, and *HRD1B* were, at least partially, regarded as a cause of inducing the constitutive

photomorphogenic phenotype of *cop1-6* (Fig. 4c and Supplementary Figs. 6c–f, 8d–g). To determine how *dcs* mutants suppress *cop1-6*, we determined the *PIF4*, *RVE1*, *ABA3*, and *HRD1B* SI values in *dcs cop1-6* double mutants. The SI values for *PIF4*, *RVE1*, *ABA3*, and *HRD1B* in dark-grown *dcs cop1-6* double mutants were partially restored to the WT_D level compared with those in *cop1-6* (Supplementary Fig. 11). These results showed that the suppressors partially rescued the AS pattern in the *cop1-6* mutant, consistent with their recovered phenotype with elongated hypocotyls.

## DCS1 physically interacts with COP1 and undergoes COP1-mediated ubiquitination and degradation

To investigate the functional relationships between COP1 and DCS proteins, we carried out a yeast two-hybrid assay to screen possible interactions between the six DCS proteins and COP1. In yeast cells, only DCS1 interacted with COP1 (Fig. 5a). Domain-mapping assays showed that COP1 interacted with the full-length DCS1 but did not interact with either N- or C-terminal parts of DCS1 (Fig. 5b, c). This suggests that the overall structure of DCS1 is necessary for its interaction with COP1. Further, by using various COP1 truncated fragments, we found that the

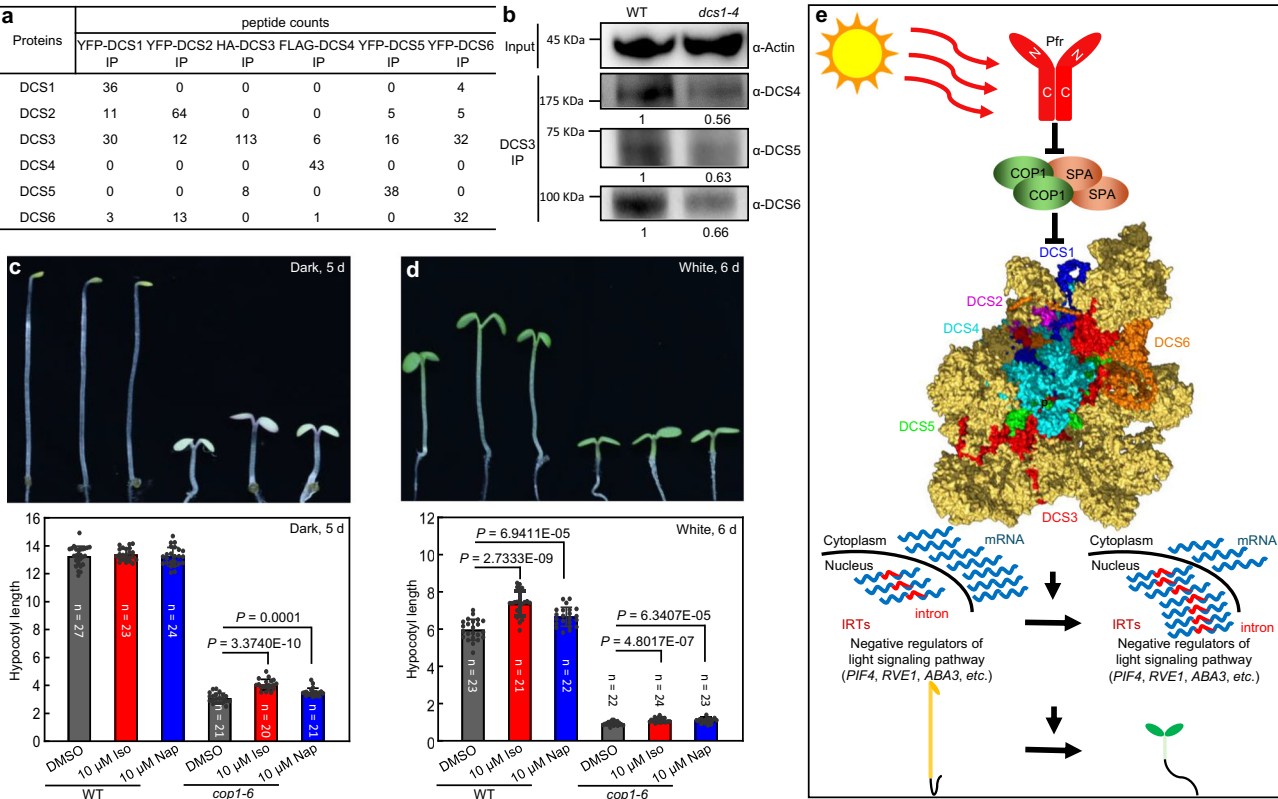

**Fig. 6 | An intact and active spliceosome is essential for light modulation of seedling photomorphogenic development. a** Interactomes of DCS proteins. Total proteins were extracted from 7-d-old white light-grown *DCS* overexpression lines. Tag-fused DCS proteins were used as baits for immunoprecipitation. The immunoprecipitates were then analyzed using mass spectrometry. **b** DCS3-associated proteins identified in WT and *dcs1-4* seedlings by Co-IP. DCS3 and DCS3-associated proteins were immunoprecipitated using an anti-DCS3 antibody, and immunoblotting was then performed with anti-DCS4, anti-DCS5, and anti-DCS6 antibodies. Actin served as a loading control. Spliceosome inhibitor treatments suppressed the *cop1-6* phenotype under both dark (**c**) and white light (9.56 μmol m$^{-2}$ s$^{-1}$) conditions (**d**). Iso isoginkgetin, Nap naphthazarin. Values are mean ± SEM. Significance is evaluated by the two-sided Student's $t$ test and $P$ values are indicated. **e** Proposed working model. Light, through COP1 modulation of the spliceosome, promotes nuclear detainment of the IRTs of negative regulators in the light signaling pathway to facilitate seedling photomorphogenesis. The structure of the DCS protein complex was generated using PyMOL v2.5 (Schrödinger, LLC). Source data are provided as a Source Data file.

coiled-coil domain of COP1 was required for its interaction with DCS1 (Fig. 5d, e).

We performed a bimolecular fluorescence complementation (BiFC) assay to confirm the interaction between DCS1 and COP1. Strong YFP fluorescent signals were clearly observed in epithelial cells of *N. benthamiana* leaves transiently co-transfected with constructs expressing DCS1, fused with the C terminus of YFP (DCS1-YFP$^C$), and COP1, fused with the N terminus of YFP (COP1-YFP$^N$), while no YFP signals were detected in the control samples (Fig. 5f). We further performed co-IP assays using *YFP-DCS1* WT transgenic plants (line #4) and the monoclonal YFP antibody, and YFP-DCS1 clearly co-immunoprecipitated endogenous COP1 (Fig. 5g). Together, these data support that DCS1 directly interacts with COP1 in vivo.

Given that DCS1 interacts with COP1 and COP1 is an E3 ligase, we investigated whether COP1 modulates DCS1 ubiquitination and protein abundance. An in vitro protein ubiquitination assay showed that the COP1-SPA4 complex ubiquitinated DCS1 in the presence of E1, E2, and ubiquitin (Fig. 5h). YFP-DCS1 protein levels were significantly increased in both *cop1-4* and *cop1-6* backgrounds compared with that in the WT background (Fig. 5i). This suggests that COP1 negatively regulates the stability of DCS1. Further, MG132 treatment markedly increased the YFP-DCS1 protein level (Fig. 5j), which suggests that DCS1 undergoes 26S proteasome-mediated degradation. Additionally, the abundance of YFP-DCS1 was gradually reduced in seedlings grown in white light upon transfer to darkness at various time points (1, 3, 6 h) (Fig. 5k), consistent with increased COP1 activity in the dark.

## An intact and fully functional spliceosome is essential for light modulation of seedling photomorphogenic development

Based on the above findings, we hypothesized that all of the DCS proteins were required components in the process of spliceosome formation and function during photomorphogenesis. They are likely components of the same spliceosome complex at some point in the spliceosome function. Intriguingly, DCS3 might be the connecting point in the spliceosome for the other five DCS proteins, as DCS3 can be immunoprecipitated by every other DCS protein (Fig. 6a). DCS3 is highly conserved across different species except for the N-terminus[64]. Prp8, the counterpart of *Arabidopsis* DCS3, which is at the heart of the spliceosome, plays a pivotal role in the activation of the spliceosome and the formation of the catalytic core for the two RNA splicing steps[49,64]. Based on the recently reported atomic structure of yeast Prp8, the mutated amino acids of the *dcs3-2* and *dcs3-3* alleles are on the outer surface of Prp8/DCS3, which may affect the interactions of DCS3 with surrounding components within the spliceosome (Supplementary Fig. 12)[64]. Moreover, we predicted the protein structures of DCS3, DCS3-1, DCS3-2, and DCS3-3 using SWISS-MODEL, a fully automated protein structure homology-modeling server. The results showed that the mutations in *dcs3-2* and *dcs3-3*, but not *dcs3-1*, bring changes to the protein structure of DCS3 (Supplementary Fig. 13). These structural changes might affect the interactions of DCS3 with surrounding components within the spliceosome.

Considering that the stability of DCS1 is controlled by light via COP1, we explored the potential alterations in the spliceosome

composition in the *dcs1-4* mutant, which is a T-DNA insertion mutant (Supplementary Fig. 14). Total proteins were extracted from WT and *dcs1-4* plants, and immunoprecipitated with an anti-DCS3 antibody, and the DCS3-associated DCS proteins were characterized using immunoblotting. The protein levels of DCS4, DCS5, and DCS6 that can be pulled down by DCS3 were significantly reduced in the *dcs1-4* mutant (Fig. 6b). Hence, DCS1 appears to be important for spliceosome association with other DCS proteins.

To obtain further support for a role of the spliceosome in seedling photomorphogenic development, we treated dark-grown WT and *cop1-6* seedlings with splicing inhibitors isoginkgetin and naphthazarin[65]. We noted that both inhibitors partially suppressed the *cop1-6* phenotype but had no effect on the WT in the dark (Fig. 6c). Splicing inhibitor treatment also promoted the hypocotyl growth of both WT and *cop1-6* seedlings in white light (Fig. 6d). These results support the conclusion that the spliceosome plays an essential role in seedling photomorphogenic development.

## Discussion

In addition to inducing alterations in the expression levels of thousands of genes, light also evokes changes in the AS patterns of numerous genes[3]. AS has been proven to play an essential role in seedling photomorphogenic development[3,26,28,30,32,36]. COP1, whose activity and nuclear abundance are tightly controlled by light, is a major light signaling suppressor[37]. In this study, our data clearly showed that COP1 is involved in light-regulated AS, mainly IR, to modulate seedling photomorphogenesis (Figs. 2 and 3). IR leads to the nuclear detainment of IRTs as storage (Fig. 3a and Supplementary Fig. 8b, c)[5,8–10]. Once the external stimulus and developmental phase change, IRTs can be fully spliced and translocated into the cytoplasm for translation[10,11]. IR is the most common AS event in *Arabidopsis*, and it could potentially be induced by various environmental factors and stresses[3,66,67]. IR-induced nuclear detainment of IRTs may play a significant role in gene expression regulation. Without transcribing mRNA de novo, this regulatory mechanism may be a quicker response to surrounding changes and the developmental phase transition. Our data clearly indicate that IR-induced nuclear detainment of IRTs represents a potential layer of gene expression regulation in plants.

The spliceosome, acting like delicate machinery, is a huge ribonucleoprotein complex. The assembly of spliceosomes on every intron is in a stepwise manner and tightly regulated[68]. To ensure fidelity and flexibility in pre-mRNA splicing, both the conformation and composition of the spliceosome are highly dynamic and under strict and precise control in all organisms[19]. In recent years, published data have indicated that post-translational modifications, including ubiquitination, methylation, phosphorylation, and more, contribute to splicing fidelity and flexibility by reversibly adjusting and fine-tuning the activity, stability, localization, interaction, folding, and assembly of spliceosomal proteins[19,48,69–72]. In the process of B^act complex formation, the U4 snRNP protein Prp3 ubiquitination, mediated by the NTC component Prp19, facilitates U4/U6-U5 triple snRNP reassembly to the spliceosome by increasing Prp3's affinity to Prp8[70,72]. Subsequently, Usp4-mediated de-ubiquitination of Prp3 decreases its affinity for Prp8, resulting in U4 departure and formation of the catalytic core[70,72]. Recent reports have suggested that dysfunction of splicing factors, such as RRC1, SFPS, and SWAP1, triggers impaired photomorphogenic responses in *Arabidopsis*[26,28,30]. Until now, it has remained largely unknown how light signals are transduced to the spliceosome to alter its activity and regulate photomorphogenic responses.

In this study, our data clearly indicate that light-induced genome-wide changes in IR events through modulation of spliceosome function give rise to alterations in nuclear-detained IRTs and thus induce the expression changes of those target genes to fine-tune seedling photomorphogenic development (Fig. 4). COP1 directly interacts with the plant-specific spliceosomal factor DCS1 and mediates its ubiquitination

and degradation (Fig. 5). Downregulation of *DCS1* caused a defect in the spliceosome functional state or recruitment of a subset of components (Fig. 6b). Therefore, through ubiquitination and degradation of DCS1 by COP1, light can modulate the activity of spliceosomes and fine-tune seedling photomorphogenesis (Figs. 5 and 6). Considering that DCS1 is the connection between COP1 and spliceosome activity and DCS1 interacts with COP1, we checked the interactions between COP1 and DCS1-1, DCS1-2, DCS1-3 proteins using yeast-two-hybrid. The results showed that the mutated proteins DCS1-1, DCS1-2, and DCS1-3 did not interact with COP1 in yeast cells (Supplementary Fig. 15a). Moreover, DCS1, but not DCS1-1, DCS1-2, and DCS1-3, interacted with DCS5 (Supplementary Fig. 15b). These might affect the spliceosome conformation/ activity. However, the mutations in DCS1 did not disrupt the interaction between DCS1 proteins and splicing factors SKIP, SR45, and U1-70K (Supplementary Fig. 15c). Therefore, it is speculated that alterations in spliceosome conformation/activity might contribute to suppress the *cop1-6* phenotype. Taken together, we propose that light contributes to the modulation of the activity or stability of the spliceosome, at least partially through COP1-mediated ubiquitination and degradation of DCS1, to regulate seedling photomorphogenic development (Fig. 6e).

## Methods

### Plant materials

The *Arabidopsis thaliana* mutants *cop1-6*[41], *pif4-2*[73], *rve1-2* (SAIL_326_A01)[74], *aba3-3* (this study), and *dcs1-4* (SALK_116275) (this study) all have the Col-0 (wide-type, WT) background. The *aba3-3* mutant, containing a large DNA fragment deletion from C1049 to A2892 of the genomic sequence of *ABA3*, was created using the CRISPR/Cas9 technique as previously described[75]. The *PIF4-YFP* WT and *myc-RVE1* WT overexpression lines all have the Col-0 (wide-type, WT) background[76,77].

### Splicing inhibitor treatment

For splicing inhibitor treatment, surface-sterilized seeds were plated on MS medium containing 10 µM isoginkgetin or 10 µM naphthazarin, while MS medium containing DMSO was used as a control. These plates were kept at 4 °C in the dark for 3 d. To induce uniform germination, all of the seeds were exposed to white light for 8 h at 22 °C and then transferred to constant dark or white light conditions in light-emitting diode growth chambers at 22 °C before phenotype analysis.

### Transgenes

To generate *ABA3* overexpression lines in the WT background, the full-length coding sequences of *ABA3* were cloned into the *pDONR-223* vector (Invitrogen) and then introduced into the plant binary vector *pEarlyGate 104* under the 35S promoter.

To generate *DCS1–DCS6* overexpression lines in the WT background, the full-length *DCS1*, *DCS2*, and *DCS5* coding sequences and the genomic sequence of *DCS6* were cloned into the *pDONR-223* vector (Invitrogen) and then introduced into the plant binary vector *pEarly-Gate 104* under the 35S promoter. *UBQ10:DCS3-HA*[48] and *UBQ10:Flag-DCS4* transgenes were introduced into the WT background.

### Construction of plasmids

The full-length coding sequences of *COP1* or *DCS1* were introduced into *pSPYNE* or *pSPYCE*, respectively, in order to produce constructs for BiFC assays[78]. The N-terminal 291 bp and full-length coding sequences of *DCS1* were cloned into the EcoRI/XhoI restriction sites of the *pET28a* vector to produce constructs for protein expression. The primers used for plasmid construction are listed in Supplementary Table 3.

### Measurement of hypocotyl length

To measure hypocotyl length of *Arabidopsis* seedlings, surface-sterilized seeds treated with 20% commercial Clorox bleach for

10 min were sown on MS plates at 4 °C in the dark for 3 d. To induce uniform germination, all seeds were exposed to white light for 8 h at 22 °C and then transferred to constant dark or white light conditions in light-emitting diode growth chambers (Percival Scientific) at 22 °C. Five-day-old seedlings were photographed using a camera (EOS80D; Canon), and the hypocotyl length was measured using ImageJ software.

## Extraction of total RNA, cytoplasmic RNA, and nuclei RNA
Subcellular fractions were isolated using a previously reported method with minor modifications[10]. White light-exposed WT, dark-exposed WT, and dark-exposed *cop1-6* were grown on MS plates at 22 °C for 5 d before collection. 3 g seedlings were ground into fine powder with liquid nitrogen, and then resuspended in 10 ml ice-cold Honda buffer (0.44 M sucrose, 1.25% (w/v) Ficoll, 2.5% (w/v) dextran T40, 20 mM HEPES-KOH pH 7.4, 10 mM $MgCl_2$, 0.15% (w/v) Triton X-100, 1 mM dithiothreitol (DTT), 1× Complete Protease Inhibitor Mixture (Roche) and 100 ng μl$^{-1}$ tRNA). The lysate was filtered through two layers of Miracloth. A 500 μl sample was collected as the total fraction, and 5 μl RNase inhibitor was added. The flowthrough was centrifuged at 3,500 rpm for 10 min at 4 °C. We then obtained the supernatant and pellet. From the supernatant, a 500 μl sample was taken and centrifuged twice in a 1.5 ml tube at 14,000 rpm for 10 min at 4 °C. The supernatant was collected as the cytosolic fraction, and 5 μl RNase inhibitor was added to it. The pellet was washed 4 times with 10 ml ice-cold Honda buffer (0.44 M sucrose, 1.25% (w/v) Ficoll, 2.5% (w/v) dextran T40, 20 mM HEPES-KOH pH 7.4, 10 mM $MgCl_2$, 0.5% (w/v) Triton X-100, 1 mM DTT, 1× Complete Protease Inhibitor Mixture (Roche), and 100 ng μl$^{-1}$ tRNA). After centrifugation at 3,500 rpm for 5 min at 4 °C, the supernatant was discarded, and the pellet was collected as the nuclei fraction.

For RNA extraction, an RNAprep Pure Plant Plus Kit (Polysaccharides & Polyphenolics-rich) (TIANGEN) was used according to the manufacturer's protocol. The RNA samples were quantified using NanoDrop-2000 (Thermo Fisher Scientific).

Western blot was applied to verify the purity of each fraction. Primary antibody anti-UGPase (Agrisera, AS05086) was used for cytoplasmic fraction-specific marker, and anti-Histone H3 (ABclonal, A2348) was used for nuclei fraction-specific marker. The antibodies were used at the dilution of 1:2000.

## Genetics screen, identification, and characterization of DCS1−DCS6
We carried out genetic screening, identification, and characterization as previously described[42]. Homozygous *dcs cop1-6* mutants were crossed with WT plants, and segregation in the F2 generations was analyzed in the dark to distinguish between intragenic and extragenic suppressors. Meanwhile, the suppressor mutants were backcrossed with *cop1-6*. F1 generation showing the WT phenotype with long hypocotyl and the segregation patterns in the F2 generations (suppressor phenotype: *cop1-6* phenotype ≈ 3:1) grown in the dark confirmed that the suppression phenotype is caused by a monogenic dominant mutation in the suppressors.

## Map-based cloning of DCS1−DCS6
Rough mapping was performed as previously described[42]. Homozygous suppressor mutants were crossed with Landsberg containing a *cop1-6* mutation to generate the mapping population. F2 generation seeds were grown in the dark at 22 °C for 3 d before phenotype investigation. The seedlings displaying the *cop1-6* phenotype with short hypocotyls were selected and transferred to white light for an additional 10 days before being selected for genomic DNA extraction and mapping. The mapping markers were designed according to the *Arabidopsis* Mapping Platform (http://amp.genomics.org.cn).

## SOLiD sequencing and mutation identification
SOLiD sequencing, as described previously, was applied to identify mutations in the suppressor mutants[42]. The SOLiD fragment library was generated according to the manufacturer's protocol (Life Technologies) using the genomic DNA of the suppressor mutants and sequenced using an SOLiD5500 sequencer. Sequencing reads were mapped to the *Arabidopsis* reference genome (TAIR10), and single nucleotide polymorphism (SNP) calling was accomplished using LifeScope version 2.5. SNPs were then sorted into four categories: EMS-induced homozygous, EMS-induced heterozygous, other homozygous, and other heterozygous. Candidate homozygous EMS-induced SNPs were identified in windows with reduced heterozygosity in the regions identified by physical mapping using in-horse perl scripts developed by Novogene Co. Ltd. (Tianjing, China).

## RNA-seq processing: mapping, differential expression analysis, and splicing analysis
For RNA-seq, three biological replicates were independently prepared from the induction of seed germination to the preparation of mRNA-seq libraries. The libraries constructed by a NEBNext RNA Library Prep Kit (NEB #E7530S/L) were sequenced on the Illumina Novaseq 6000 platform by Novogene Co. Ltd. (Tianjing, China), and 150 bp paired-end reads were generated. Clean reads were obtained by removing reads containing adapters, reads containing ploy-N, and low-quality reads from the raw data. At the same time, the Q20, Q30, and GC content of the clean data were calculated. RNA-seq reads for every replicate were aligned against the *Arabidopsis* reference sequence (TAIR10) using Hisat2 v2.0.5. The mapped reads of every sample were assembled using StringTie (v1.3.3b) in a reference-based approach to predict novel transcripts. FeatureCounts v1.5.0-p3 was employed to count the number of reads mapped to every gene, and the Transcripts Per Million (TPM) of every gene was calculated based on the length of the gene and read count mapped to that gene. Differential expression analysis of two conditions with three biological replicates was performed using the DESeq2 R package (v1.20.0). Genes with an adjusted FDR ≤ 0.05 and a fold change ≥2 found by DESeq2 were assigned as differentially expressed. For the splicing analysis, the same alignment files generated by Hisat2 and annotation files generated by StringTie were used as input for rMATS (v4.1.2) based on the Junction Counts (JC) method to test for differentially spliced transcripts. Alternative splicing events were identified by rMATS (v4.1.2) and the sites with $P < 0.05$ between compared pairs were assigned as differentially spliced sites.

## Production of polyclonal antibodies
To generate a DCS1-specific antibody, the N-terminal 97 amino acids of DCS1 were cloned into *pET28a*. The recombinant His-DCS1 protein was expressed in *E. coli* strain BL21 (DE3). Polyclonal DCS1 antibodies were raised by immunizing rabbits using purified His-DCS1 recombinant protein as an antigen. Polyclonal DCS3, DCS4, DCS5, and DCS6 antibodies were raised by immunizing rabbits using synthetic peptides as antigens. The 70−89 amino acids of DCS3, the 1773−1787 amino acids of DCS4, the 66−80 amino acids of DCS5, and the 9−26 amino acids of DCS6 were all synthesized in vitro.

## Quantitative real-time PCR
cDNAs were synthesized from 2 μg of total RNA, cytoplasmic RNA, and nuclear RNA using the 5X All-In-One RT Master Mix cDNA synthesis system (Applied Biological Materials) according to the manufacturer's protocol. Quantitative real-time PCR was performed using the StepOnePlus Real-Time PCR Detection System (Applied Biosystems) and Hieff® qPCR SYBR Green Master Mix (High Rox Plus) (Yeasen). The housekeeping gene *PROTEIN PHOSPHATASE 2A* (*PP2A*) was used as the reference gene. The primers used in this study are listed in Supplementary Table 3.

## Bimolecular fluorescence complementation assay (BiFC)

The constructs *pSPYNE-35S-COP1* and *pSPYCE-35S-DCS1* were introduced into *Agrobacterium* strain GV3101 using the freeze–thaw method. Then, the indicated transformant pairs in infiltration buffer (10 mM MgCl$_2$, 150 mM acetosyringone, and 10 mM MES, pH 5.6) were infiltrated into *Nicotiana benthamiana* leaves. YFP fluorescence signals were detected using a confocal laser scanning microscope (LSM510 Meta; Carl Zeiss) 48 h after infiltration.

## Co-IP assay

For the Co-IP assays, 7-d-old white light-grown WT and *YFP-DCS1* WT #4 seedlings were transferred to the dark for 16 h before harvesting. Plant seedlings (1.5 g) were ground into powder with liquid nitrogen and resuspended in 800 μl protein extraction buffer (50 mM Tris-HCl, pH 7.5, 100 mM NaCl, 1 mM EDTA, 10% (v/v) glycerol, 0.2% Triton X-100, 1 mM PMSF, and 1× Complete Protease Inhibitor Mixture (Roche)). After centrifugation at 14,000 rpm for 10 min three times, the supernatants were transferred into new 1.5 ml tubes. The protein concentration was determined using Quick Start Bradford 1× Dye Reagent (Bio-Rad) according to the manufacturer's protocol. Equal amounts of total proteins were incubated with Anti-GFP Nanobody Agarose Beads (Alpla Life) for 6 h at 4 °C. After centrifugation (3000 rpm for 2 min at 4 °C) and washing four times with protein extraction buffer, the precipitates in 1× SDS protein loading buffer were boiled for 10 min at 100 °C before SDS-PAGE gel electrophoresis and immunoblot analysis.

## LC-MS/MS analysis

LC-MS/MS analysis was conducted according to previously described methods with some modifications[79]. In brief, the bound proteins were digested on-bead with 300 μl of 50 mM NH$_4$HCO$_3$ containing 2 μg of Lys-C/Trypsin protease mix (Promega, Madison, WI) overnight at 37 °C. All the peptides were purified using StageTips prior to LC MS/MS analysis. The concentration of peptides was determined using the BCA assay.

2 μg peptides were resuspended in 0.1% (v/v) formic acid solution and then analyzed using an Orbitrap Eclipse mass spectrometer (Thermo Fisher Scientific). The peptides were separated by Easy-nLC 1200 (Thermo Fisher Scientific), which was equipped with a 350 mm in-house C18 analytical column. The peptide mixture was separated on the analytical column with a liquid gradient from 3–40% of solvent B (80% acetonitrile/0.1% FA) at a flow rate of 300 nl/min for 75 min. The parameters for a full MS survey scan were set as follows: a resolution of 60,000 at 400 m/z over the m/z range of 300–1500, Automatic gain controls (AGC) target of 4E5. The parent ions were selected by data-dependent MS/MS mode with a 0.7 sec cycle and fragmented by high-energy collision-induced dissociation (HCD). For MS/MS detection by IonTrap, the scan rate was set at Turbo, the AGC target value was 1E4, and the maximum injection time was 20 ms. Dynamic exclusion was enabled for 20 s.

The raw files were processed using Proteome Discoverer software (Thermo Fisher Scientific, version 2.4) for peptide identification and searched against the *Arabidopsis* Information Resource (TAIR 10). The following parameters were used: oxidation of Met was set as variable modifications, and a maximum of two missed cleavages was allowed. The false discovery rates of the peptides were set at 1% FDR.

## Western blot and antibodies

Total proteins were extracted from *Arabidopsis* seedlings using the denatured protein extraction buffer containing 100 mM NaH$_2$PO$_4$, 10 mM Tris·HCl (pH 8.0), 200 mM NaCl, 8 M urea, 1 mM PMSF, and 1× complete protease inhibitor cocktail (Roche). The protein samples were denatured by boiling at 100 °C for 10 min. Equal amounts of total proteins were separated in 10% SDS-PAGE gels and then transferred onto PVDF membranes. The primary antibodies used in this study were anti-DCS1 (this study), anti-DCS4 (this study), anti-DCS5 (this study), anti-DCS6 (this study), anti-COP1 (McNellis et al.[41]), anti-GFP (Abmart, M20004M), and anti-Actin (Sigma-Aldrich, A0480). Anti-DCS1, anti-DCS4, anti-DCS5, anti-DCS6, and anti-COP1 antibodies were used at the dilution of 1:1000. Anti-GFP and anti-Actin antibodies were used at the dilution of 1:2000.

## RNA FISH

RNA FISH was performed as previously described, with some modifications[80]. Seven-day-old WT seedlings were fixed in 1:1 fixation buffer (120 mM NaCl, 7 mM Na$_2$HPO$_4$, 3 mM NaH$_2$PO$_4$ (pH 7.5), 2.7 mM KCl, 0.1% Tween-20, 80 mM EGTA (pH 8.0), 5% (wt/vol) formaldehyde, 10% (vol/vol) DMSO): heptane for 30 min at room temperature with shaking and then dehydrated in 100% (vol/vol) methanol twice for 5 min each and dehydrated in 100% (vol/vol) ethanol three times for 5 min each. Subsequently, the seedlings were incubated for 30 min in 1:1 ethanol: xylene at room temperature, dehydrated in 100% (vol/vol) ethanol twice for 5 min each, and dehydrated in 100% (vol/vol) methanol twice for 5 min each. Then, the seedlings were postfixed in 1:1 methanol: fixation buffer without formaldehyde for 30 min at room temperature, followed by two rinses in fixation buffer without formaldehyde. Before hybridization, the seedlings were rinsed in 1 ml of PerfectHyb Plus hybridization buffer (Sigma-Aldrich; H-7033) and then prehybridized in 1 ml of hybridization buffer for 1 h at 52 °C. After that, 6 μl of 1 μM 5′-Cy3-labeled probes was added into the hybridization buffer, and samples were incubated in the dark at 52 °C for 12 h. After hybridization, the samples were washed twice for 60 min each in 2× SSC (300 mM NaCl, 30 mM sodium citrate (pH 7.0)) containing 0.1% SDS, followed by a 20 min wash in 0.2× SSC containing 0.1% SDS at 50 °C. The samples were then incubated in 0.2× SSC containing 2 μg/ml DAPI at 37 °C for 15 min in the dark. After being washed twice with 0.2× SSC, the samples were stored in 0.2× SSC until they were imaged. The probes used in this study are listed in Supplementary Table 3.

## Ortholog identification and phylogenetic construction

A total of 31 plant species covering the major diversity of plants, 2 fungi (*Saccharomyces cerevisiae* and *Neurospora crassa*), and 3 animals (*Drosophila melanogaster*, *Mus musculus*, and *Homo sapiens*) were screened for DCS orthologs. The genomic sequences were retrieved from public databases (Supplementary Table 1), with the primary sources being NCBI, Phytozome v13 (https://phytozome-next.jgi.doe.gov/), and EnsemblPlants (http://plants.ensembl.org/index.html).

The *Arabidopsis* DCS1 (AT5G53800) protein sequence was used as a query to perform BLASTP searches against all of the genes annotated in 36 representative genomes with an E-value threshold of 1e−3. Domain annotation was performed with InterPro (v89.0). Similar protein sequences with "Coil" domain annotation were considered the orthologs of DCS1 and then aligned with the L-INS-I strategy in MAFFT (v7.475); the positions with more than 50% gaps were removed using Phyutility (v2.2.6). The best-fit substitution model was chosen by ModelFinder. Subsequently, the maximum likelihood method implemented in IQ-TREE (v2.1.4-beta) was used to construct the phylogenetic tree of DCS1 with 2000 bootstrap replicates, and the extremely long branches were removed manually. Some genes contained multiple transcript isoforms due to alternative splicing; therefore, the representative protein, which was the longest one, was selected. The phylogenetic tree was annotated using iTOL (similarly, hereinafter).

The *Arabidopsis* DCS4 (AT4G01020, AtDEAH11 of the DEAH/RHA family) protein sequence was used as a query to perform a BLASTP search. All of the sequences with an E-value of less than 1e−3 and with the HMMER annotation of "DEAD" were aligned using MAFFT and trimmed using Phyutility as described above. Sequences belonging to the monophyletic group containing the *Arabidopsis* DCS4 were identified as orthologs. Finally, they were realigned and trimmed to reconstruct the DCS4 phylogenetic tree using IQ-TREE with 2000 bootstrapping events.

## Statistical analysis

Statistical analyses were performed using Microsoft Excel and Graph-Pad Prism version 5.0.

## Reporting summary

Further information on research design is available in the Nature Portfolio Reporting Summary linked to this article.

## Data availability

All materials in this study are available from the corresponding author upon request. RNA-seq data have been deposited in the NCBI GEO under accession number PRJNA880452. The mass spectrometry proteomics data have been deposited to the ProteomeXchange Consortium via the PRIDE partner repository with the dataset identifier PXD043190. *Arabidopsis* reference genome (TAIR10) was used in this study. Source data are provided with this paper.

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

## Acknowledgements

We thank Dr. Yuping Qiu (Southern University of Science and Technology) for the help on the operation of confocal microscope. We thank Prof. Rongcheng Lin (Chinese Academy of Sciences) for providing genetic materials *rve1-2* and *myc-RVE1* WT. We thank Prof. Xiaofeng Cao (Chinese Academy of Sciences) for providing plasmid *UBQ10-DCS3-HA*. We thank Prof. Ping Yin (Huazhong Agricultural University) for providing recombinant protein COP1-SPA4. This work was supported by the National Natural Science Foundation of China (NSFC) Key Program (32230006), Southern University of Science and Technology (Y01226026), Shenzhen Science and Technology Program (ZDSYS202230626091659010), Key Laboratory of Molecular Design for Plant Cell Factory of Guangdong Higher Education Institute (2019KSYS006), and the National Key R&D Program of China (2017YFA0503800) to X.W.D., and the National Natural Science Foundation of China (32100198) to H.Z.

## Author contributions

H.Z. and X.W.D. conceived the project and designed the experiments. H.Z., T.Y., S.C., Y.F., G.Q., X.Z., Y.H., J.L., F.L., and D.X. conducted the experiments. H.Y.Z. performed bioinformatic analysis. X.W.D., N.W., and H.Z. analyzed the data. X.W.D., N.W., and H.Z. wrote the article.

## Competing interests

The authors declare no competing interests.
