## [Peer Review file · Nature Communications]

REVIEWER COMMENTS

Reviewer #1 (Remarks to the Author):

The manuscript by Zhou et al. described findings explaining why *cop1-6* has a photomorphogenic phenotype in the dark in addition to having stabilized positive regulators of photomorphogenesis without the full capacity of COP1. By screening and analyzing genetic suppressors of *cop1-6*, the authors identified multiple dominant alleles mapped to various components of spliceosomes. The defects in spliceosome may impact alternative splicing, particularly IR in plants, which were further investigated for selected genes and correlated with their nuclear retention and phenotypic contributions. Overall the approaches were solid and robust. The findings provide new mechanistic insights into light-regulated gene expression regulation involving COP1 in modulating spliceosomes. Before this manuscript is considered for publication, a few major points are listed below for the editor and authors to consider.

1. *dcs2-1*, *2-2*, *2-3* and *2-4* have comparable hypocotyl length but quite distinct differences in the degree of cotyledon opening. Can the authors provide some discussion or speculation based on the protein sequences/annotated motifs for the mutation sites? Also, if all DCSs are spliceosome components, are there possible explanations for the different degrees of suppression shown in Fig. 1a, considering all but one mutants carry missense mutations? Do the mutations compromise the protein levels, spliceosome conformation/complex/activity to different degrees, or impact target mRNAs differently?
2. The light/COP1-induced or -repressed IR appeared only to constitute a very small portion of the total transcripts (Fig. 2f). Also, other than PIF4, the splicing indexes for the other selected genes are relatively low (Supplementary Fig 5, different SI in the nucleus in Supplementary Fig 6, possibly due to image quantitation?). This raises a concern about whether the nuclear retention of these IRTs plays a crucial role in regulating the gene expression levels by keeping them from being translated into the cytosol. Consistent with this notion, mutations in these genes (dramatically reduced expression of the selected genes) only resulted in minor phenotypic changes (Fig. 4a) compared to the overexpression lines. Perhaps some discussion on nuclear retention of “many light/COP1-regulated IRTs” and their “collective expression regulation” mark the impact on photomorphogenesis?
3. The methods, data processing, and presentation for the RNAseq data shown in Figures 2 and 3 can be improved for better clarity. For example, it’s unclear how many biological replicates were conducted. What are Samples 1/2 and replicates shown in Supplemental Data 1? The terminologies should be better explained for general readers, e.g., IncLevel in Fig. 3a, the up-IR, or the down-IR in pairwise comparisons of the samples marked. Fig. 3a was used to claim that greater than 90% of the IRTs are higher/retained in the nucleus. The separation of up- and down-IR complicated the conclusion. Perhaps a simple scatter plot with N and C read distribution for all three samples is straightforward enough for making the nuclear retention point?

4. DCS1 interacts with COP1, and COP1 tags DCS1 for degradation to regulate its protein abundance (and perhaps spliceosome level/capacity) in the dark. Whether the authors check if the missense mutations in DCS1 affect the interaction between DCS1 and COP1 to explain the suppression of the weak allele *cop1-6*? Or do the DCS1 mutations impair the interaction of DCS1 with the other spliceosome components?

Minor points:

1. Spelling and grammar checks are needed throughout the manuscript.
2. Lines 60-61, “Dysfunctional mRNA splicing is always linked to cancers....”. Is it true is “always” linked to cancers?
3. Some of the statements could be toned down. For example, in lines 114-116, “All the dominant mutants carried a missense mutation in their protein sequences, except a nonsense mutation in *dcs1-3* (Fig. 1b, c), indicating that the mutations in DCS genes can suppress the phenotype of the *cop1-6* mutant.” The first half of the sentence doesn’t justify the statement in the 2nd half.

Reviewer #2 (Remarks to the Author):

The authors isolate dominant repressors of the *cop1-6* mutant and discover that 6 of these genes have roles in the spliceosome, which promotes and coordinates the process of pre-mRNA splicing. They perform several genetic, RNA-sequencing, interaction assays (IP-MS) and ubiquitination assays to support the interaction between COP1 and the DCS mutants (mostly DCS1 which is a plant specific gene). This makes this an interesting and significant paper that describes a link between the spliceosome to light regulation of splicing, photomorphogenesis and protein turnover via COP1.

My comments, concerns and questions described below mostly relate to the intron retention (IR) part of the paper. The main drive and title of the paper is that COP1 may have a role in light dependent nuclear retention of intron containing transcripts (IR transcripts) affecting photomorphogenesis. I have less confidence in this conclusion based on the results presented here, which is largely based on numbers of overlapping alternatively spliced (AS) genes between wild and *cop1-6* and through nuclear versus cytoplasmic AS gene numbers. The authors do show 4 genes as examples, but I don’t think they are particularly strong AS examples, as the differences between IR and the fully spliced product is quite limited. If this is being related to function, then the levels of the functional transcript need to be considered as well. I also think more detailed study of different examples is needed.

In addition, the nuclear v cytoplasmic data presented is an interesting data set that could be studied in far more detail in a splicing sense than described here. What genes make it out compared to those that don't? Why the large difference in numbers? Does the large change in AS numbers of genes mean that most transcripts that are transported to the cytoplasm and translation are single transcripts and very few AS gene transcripts make it to translation and alternative proteins made?

In Fig 3 d and e. IRTs are more abundant in the nuclear fraction, but so are the other AS events. Can the authors comment on the ~10 fold less SE, Alt 5 and 3'SS in the cytoplasmic fraction (~20 fold less for IRs). Are these events also largely retained in the nucleus? This is also related to picking out the IRs in Ln 205 as being 4% in the cytoplasmic. As an example of an alternative group of AS genes, SEs are also very low compared to the nuclear fraction; about 11% cytoplasmic to the nuclear total. All alternative splicing needs to be considered.

The authors suggest a role for COP1 regulating gene expression through IR retention in the nucleus and present some evidence showing intron retained transcripts (IRT) are largely retained in the nucleus. Gene expression is under the circadian clock and will show substantial changes in gene expression between light and dark. This will also result in large changes in AS too and will generate numbers of genes showing alternative splicing including IR and the other AS events. The clock is not considered or mentioned here despite the strong evidence that other suppressors of cop1-6 are clock linked genes.

Ln 209. It is about 50% overlap in the nuclear fraction (Fig3 d) but about 30% in the cytoplasmic fraction (Fig 3e).

Fig 4 and Ln 213-230 and particularly the summary conclusion line 228-230. I do not see the connection with intron retention. This is not about intron retention and its regulation. This is functional testing of cop1-6 in other mutant backgrounds. It shows that the cop1 mutation is the dominant player when it comes to hypocotyl length that is largely not overcome by overexpression of these other genes. This section needs reviewed.

Ln 233-235. I don't think you can say this. There is still plenty spliced product in your genes and probably many more genes involved. Can you detect these genes in your cytoplasmic and nuclear RNA-seq samples? Splicing occurs on ~80% of plant genes, so alterations of splicing can have a broad effect. Do changes occur in the efficiencies and accuracies of constitutively spliced genes?

Ln 280 and Supp. Fig 7. dcs1-4 is a T-DNA insertion line, why does supplemental also describe aba3-3 here as a CRSIPR? Not sure where this fits in and be better to see a model of DCS1-4 T-DNA insertion position.

Ln 286-290. Despite the significant results reported, this can only be described as partial at best. Isoginkgetin affects splicing in a general way (preventing stable recruitment of the U4/U5/U6 tri-small nuclear ribonucleoprotein) so it could well be affecting something else that leads to these changes. In addition, the fact that they continue to live means that the level of application of the inhibitor is minimal on splicing. How much drug was used? Not described in methods. Similar argument with Naphthazarin (check spelling – Ln 287).

Ln 303. But the data indicates that it retains all forms of AS gene in the nucleus not just IR.

Ln 158-160 and Supp. Fig 4. It is very interesting that GO terms identify genes with a role in pre-mRNA in response to light and these are not present in cop1-6 affected AS genes. Further supporting a role for Cop1 in splicing and alternative splicing (AS). The authors could add this.

At a technical level.

Ln 164-169 and Fig 2. In relation to Fig 2 b,c and f., how much of the IRs reported is noise? A level of IRs may well be partially spliced random products, generally found at low expression values despite being reported as significant differences. Supp Fig 5 validations show splicing indexes for RVE1 and HRD1B as <5% of the total transcripts. How confident are the authors that the couple of percentage points differences are real responses to light and mutant – despite the significant values reported?

Ln 409. What sequencing was performed and by who. How were the libraries created. Paired-end sequencing? Result of quality assessment? What are the mapping rates?

Ln 417. The calculation and quantification values of FPKM is not used to compare between sample sets. RNA-seq quantifications are normalised first for gene length, and then sequencing depth with outputs in TPM values that are proportionately comparable between samples.

Supp. Fig 5 a and b. Despite the DNase step in the RNA extraction kit used, how sure are the authors that there is no DNA in the RNA samples used for semi quantitative RT-PCR. I would recommend designing primers across the retained intron and a second spliced intron, that way you will know

that the retained intron is not due to contaminating DNA, particularly when the amounts of RT-PCR product is proportionately lower than the spliced products.

Ln 433-439. Results presented were semi-quantitative PCR reactions run out on a gel. Where are the quantitative real time results? Quantitative data not mentioned in the text other than as a method. How many PCR cycles were used in the semi quantitative RT-PCR analysis. High PCR cycles where the amplification is outwith the exponential phase is not semi-quantitative. It is important to state this.

General questions.

It will be interesting to do an RNA-seq experiment that included at least the cop1-6, dcs1 line to determine whether the mutation of DCS1 compensates at least partially for the splicing of the genes affected by the cop1-6 mutation.

Do the repressors affect splicing of cop1-6 that may alters its function?

The authors say they identified 13 dominant repressors of cop1-6 and describe 6 of them. What do the other 7 do?

Minor.

Ln 144. Bit odd to have Fig. 6 before Fig. 2 to 5.

Response letter to reviewers' comments

REVIEWER COMMENTS

Reviewer #1 (Remarks to the Author):

The manuscript by Zhou et al. described findings explaining why *cop1-6* has a photomorphogenic phenotype in the dark in addition to having stabilized positive regulators of photomorphogenesis without the full capacity of COP1. By screening and analyzing genetic suppressors of *cop1-6*, the authors identified multiple dominant alleles mapped to various components of spliceosomes. The defects in spliceosome may impact alternative splicing, particularly IR in plants, which were further investigated for selected genes and correlated with their nuclear retention and phenotypic contributions. Overall the approaches were solid and robust. The findings provide new mechanistic insights into light-regulated gene expression regulation involving COP1 in modulating spliceosomes. Before this manuscript is considered for publication, a few major points are listed below for the editor and authors to consider.

1. *dcs2-1*, *2-2*, *2-3* and *2-4* have comparable hypocotyl length but quite distinct differences in the degree of cotyledon opening. Can the authors provide some discussion or speculation based on the protein sequences/annotated motifs for the mutation sites? Also, if all DCSs are spliceosome components, are there possible explanations for the different degrees of suppression shown in Fig. 1a, considering all but one mutants carry missense mutations? Do the mutations compromise the protein levels, spliceosome conformation/complex/activity to different degrees, or impact target mRNAs differently?

Response:

Thank you for the suggestion. We have discussed some possible explanations for the differences in the degree of cotyledon opening of *dcs2 cop1-6* mutants in the revised "Results" section. In *dcs2-1 cop1-6* and *dcs2-3 cop1-6*, Gly367 and Gly495 were mutated to Glu and Arg, respectively. Both Glu and Arg were charged amino acids. In *dcs2-2 cop1-6* and *dcs2-4 cop1-6*, Ala457 and Asp512 were mutated to uncharged amino acids (Fig. 1b). Hence, there may be some differences in the physical properties of amino acids caused the different degrees of suppression. The distribution of the mutation sites in DCS proteins is within highly conserved regions among different species.

We think that the mutations in DCS proteins, including the missense mutation, impact the protein-protein interactions, but we cannot exclude other possibilities at this point in time. DCS1 interacts with COP1, but the mutated proteins DCS1-1, DCS1-2, and DCS1-3 did not interact with COP1 in yeast cells (Fig. R1a). Therefore, the mutations in the DCS1 protein might affect protein stability (as a target of COP1 E3). Moreover,

DCS1, but not DCS1-1, DCS1-2, and DCS1-3, interacted with DCS5 (Fig. R1b). This might affect the spliceosome conformation/activity. In addition, based on the recently reported atomic structure of yeast Prp8, the mutation sites of *dcs3-2* and *dcs3-3* alleles are on the outer surface of the RT/En domain (see revised Supplemental Fig. 10), a crucial protein-binding surface that may affect interactions of DCS3 with surrounding components within the spliceosome^{1, 2}.

Fig. R1. The mutations in DCS1 affect protein–protein interactions. a Yeast two-hybrid screening for the interactions between DCS1 proteins and COP1. b Yeast two-hybrid screening for the interactions between DCS1 proteins and DCS2, DCS3-N, DCS3-C, DCS5, and DCS6. c Yeast two-hybrid screening for the interactions between DCS1 proteins and the splicing factors SKIP, SR45, and U1-70K.

2. The light/COP1-induced or -repressed IR appeared only to constitute a very small portion of the total transcripts (Fig. 2f). Also, other than PIF4, the splicing indexes for the other selected genes are relatively low (Supplementary Fig 5, different SI in the nucleus in Supplementary Fig 6, possibly due to image quantitation?). This raises a concern about whether the nuclear retention of these IRTs plays a crucial role in regulating the gene expression levels by keeping them from being translated into the cytosol. Consistent with this notion, mutations in these genes (dramatically reduced expression of the selected genes) only resulted in minor phenotypic changes (Fig. 4a)

compared to the overexpression lines. Perhaps some discussion on nuclear retention of “many light/COP1-regulated IRTs” and their “collective expression regulation” mark the impact on photomorphogenesis?

Response:

We agree that it was not described clearly in the prior manuscript. The splicing index (SI) in revised Supplementary Fig. 6 was checked using total RNA, while the SI in revised Supplementary Fig. 7 was checked using nuclear RNA. Both were checked by qRT-PCR. The intron-retained transcripts (IRTs) were thought to be detained in the nucleus, indicating that the SI in the total RNA should be lower than the SI in the nuclear RNA.

Thank you for your suggestion. According to our data, hundreds of IRTs were regulated by both light and COP1. Hence, we agree that many other light-/COP1-regulated IRTs, not just the four genes we displayed, are involved in this process. We have added some discussion in the revised “Results” section. It should also be noted that this modulation of IRTs is just part of the COP1-affected targets, not all of the light regulation effects modulated by COP1.

3. The methods, data processing, and presentation for the RNAseq data shown in Figures 2 and 3 can be improved for better clarity. For example, it’s unclear how many biological replicates were conducted. What are Samples 1/2 and replicates shown in Supplemental Data 1? The terminologies should be better explained for general readers, e.g., IncLevel in Fig. 3a, the up-IR, or the down-IR in pairwise comparisons of the samples marked. Fig. 3a was used to claim that greater than 90% of the IRTs are higher/retained in the nucleus. The separation of up- and down-IR complicated the conclusion. Perhaps a simple scatter plot with N and C read distribution for all three samples is straightforward enough for making the nuclear retention point?

Response:

Thank you for mentioning this. We have revised the manuscript in the “Methods” and “Supplemental Data 1” sections according to your suggestions. We have explained IncLevel in the revised legend of Fig. 3a.

Thank you for your suggestion to improve Fig. 3a. We have revised Fig. 3a as you suggested (see revised Fig. 3a). The revised Fig. 3a looks better and is easier to understand.

4. DCS1 interacts with COP1, and COP1 targets DCS1 for degradation to regulate its protein abundance (and perhaps spliceosome level/capacity) in the dark. Whether the authors check if the missense mutations in DCS1 affect the interaction between DCS1 and COP1 to explain the suppression of the weak allele cop1-6? Or do the DCS1 mutations impair the interaction of DCS1 with the other spliceosome components?

Response:

DCS1 interacted with COP1, but the mutated proteins DCS1-1, DCS1-2, and DCS1-3 could not interact with COP1 in yeast cells (Fig. R1a). Moreover, DCS1, but not DCS1-1, DCS1-2, and DCS1-3, interacted with DCS5 (Fig. R1b). However, the mutations in DCS1 did not disrupt the interaction between DCS1 proteins and splicing factors SKIP, SR45, and U1-70K (Fig. R1c).

Minor points:

1. Spelling and grammar checks are needed throughout the manuscript.

Response:

Thank you for your suggestion. We have sent the manuscript to a professional language editor to improve the English throughout the manuscript.

2. Lines 60-61, “Dysfunctional mRNA splicing is always linked to cancers...”. Is it true is “always” linked to cancers?

Response:

We have revised this typo.

3. Some of the statements could be toned down. For example, in lines 114-116, “All the dominant mutants carried a missense mutation in their protein sequences, except a nonsense mutation in *dcs1-3* (Fig. 1b, c), indicating that the mutations in DCS genes can suppress the phenotype of the *cop1-6* mutant.” The first half of the sentence doesn’t justify the statement in the 2nd half.

Response:

Thank you for your suggestion. We have removed the second part of this sentence.

Reviewer #2 (Remarks to the Author):

The authors isolate dominant repressors of the *cop1-6* mutant and discover that 6 of these genes have roles in the spliceosome, which promotes and coordinates the process of pre-mRNA splicing. They perform several genetic, RNA-sequencing, interaction assays (IP-MS) and ubiquitination assays to support the interaction between COP1 and the DCS mutants (mostly DCS1 which is a plant specific gene). This makes this an interesting and significant paper that describes a link between the spliceosome to light regulation of splicing, photomorphogenesis and protein turnover via COP1.

My comments, concerns and questions described below mostly relate to the intron retention (IR) part of the paper. The main drive and title of the paper is that COP1

may have a role in light dependent nuclear retention of intron containing transcripts (IR transcripts) affecting photomorphogenesis. I have less confidence in this conclusion based on the results presented here, which is largely based on numbers of overlapping alternatively spliced (AS) genes between wild and *cop1-6* and through nuclear versus cytoplasmic AS gene numbers. The authors do show 4 genes as examples, but I don't think they are particularly strong AS examples, as the differences between IR and the fully spliced product is quite limited. If this is being related to function, then the levels of the functional transcript need to be considered as well. I also think more detailed study of different examples is needed.

Response:

To better understand the number of AS events, we changed the statistical threshold of RNA-seq data analysis from $FDR < 0.05$ to $P < 0.05$. The numbers of overlapping AS events between WT_L and *cop1-6_D* were improved (see revised Figs. 2 and 3). Considering that one gene can have more than one AS event, we also investigated the number of overlapping AS genes between WT_L and *cop1-6_D*. Nearly 61% of COP1-responsive AS genes overlapped with light-responsive AS genes in our analysis (Fig. R2).

Fig. R2. Venn diagram illustrating the number of AS genes that display either light-dependent changes (red), COP1-dependent changes (green), or both light- and COP1-dependent changes (yellow). AS, $P < 0.05$.

It is interesting that there was a big difference in the amount of nuclear vs. cytoplasmic data. Previous studies have reported that light enhances the global translational efficiency in *Arabidopsis*, while COP1 represses it in the dark^{3,4}. This indicates that many more IRTs are fully spliced to remove the introns and then translocated into the cytoplasm for translation in WT_L and *cop1-6_D*, compared to that in WT_D. This may explain why nuclear vs. cytoplasmic IR event numbers were low in WT_L and *cop1-6_D*.

We checked the amount of the functional transcripts of *PIF4*, *RVE1*, *ABA3*, and *HRD1B* in the total transcripts. The proportion of *PIF4*, *RVE1*, and *ABA3* functional

transcripts was downregulated in WT_L and *cop1-6_D*, compared with WT_D (Fig. R3). While the proportion of *HRD1B* functional transcripts was higher in WT_L and *cop1-6_D* than in WT_D (Fig. R3).

Fig. R3. Light and COP1 induce changes in the functional transcripts of *PIF4* (a), *RVE1* (b), *ABA3* (c), and *HRD1B* (d). The exons are represented by yellow boxes, the UTR regions are represented by green boxes, and the introns are represented by blue lines. The red boxes represent the intron-retaining exons. mRNA-1 represents the splice variant considered to encode the functional full-length protein, while mRNA-2 represents the intron-containing splice variant identified by mRNA-seq. The values are shown as the mean \pm SE ($n = 3$). Asterisks indicate statistical significance, as determined using Student's *t*-test (* $P < 0.05$; ** $P < 0.01$).

We checked 8 additional IR events identified by mRNA-seq using semiquantitative RT-PCR analysis (Fig. R4, Supplemental Fig. 5 in the revised manuscript). The IRTs of *PAP1*, *NAC062*, *bZIP63*, *CKG*, and *FRS1* were upregulated in WT_L and *cop1-6_D*, compared with WT_D (Fig. R4). The IRTs of *bHLH23*, *PIN4*, and *BBX17* were higher in WT_D than in WT_L and *cop1-6_D* (Fig. R4). Thus, there is a widespread trend of light control of AS regulation mediated by the COP1-spliceosome.

We should note that this COP1-mediated light control of AS is only part of the COP1-mediated light control of development, as many other aspects modulated by COP1 are still true and have their own roles.

Fig. R4. Validation of the IR events identified by mRNA-seq. a Gene structures of *PAP1*, *NAC062*, *bZIP63*, *CKG*, *FRS1*, *bHLH23*, *PIN4*, and *BBX17*. Red arrows indicate the locations of the primers used to confirm the IR events of each gene through semi-quantitative RT-PCR. The exons are represented by yellow boxes, the UTR regions are represented by green boxes, and the introns are represented by blue lines. b IR event conformation using semi-quantitative RT-PCR. Red arrows indicate the positions of the IRTs. Blue arrows indicate the positions of the intron-spliced transcripts. The number of PCR cycles used in this analysis were as follows: 25 cycles for *ACT2*, 27 cycles for *PAP1*, *NAC062*, *bZIP63*, and *CKG*, 29 cycles for *FRS1*, and *PIN4*, 32 cycles for *bHLH23*, and 34 cycles for *BBX17*.

In addition, the nuclear v cytoplasmic data presented is an interesting data set that could be studied in far more detail in a splicing sense than described here. What genes make it out compared to those that don't? Why the large difference in numbers? Does the large change in AS numbers of genes mean that most transcripts that are transported to the cytoplasm and translation are single transcripts and very few AS gene transcripts make it to translation and alternative proteins made?

Response:

It is very interesting to us that there is a large difference in the amount of nuclear vs. cytoplasmic data. We discussed this above and added some discussion in the revised "Results" section.

We performed GO analysis of the IR genes derived from the nuclear vs. cytoplasmic data. We found that GO terms related to 'response to light stimulus' and 'response to light intensity' were significantly enriched in groups WT_L_N vs. WT_L_C and *cop1-6_D_N* vs. *cop1-6_D_C*, but not in group WT_D_N vs. WT_D_C (Fig. R5).

Fig. R5. GO analysis of nuclear vs. cytoplasmic changed IR genes identified by mRNA-seq. Top 30 enriched GO terms (biological process aspect) for IR-changed genes derived from groups WT_L_N vs. WT_L_C (a), WT_D_N vs. WT_D_C (b), and *cop1-6_D_N* vs. *cop1-6_D_C* (c).

According to our data, in the WT_L, WT_D, and *cop1-6_D* samples, the AS variant levels for approximately 90% of the IRTs were consistently higher in the nucleus than in the cytoplasm, and among them, > 59% of the IRTs were undetectable in the cytoplasmic fractions of all the three samples (Fig. 3a and Supplemental Data 1). We speculate that those AS gene transcripts retained in the nucleus are not transferred to the cytosol for translation.

In Fig 3 d and e. IRTs are more abundant in the nuclear fraction, but so are the other AS events. Can the authors comment on the ~10 fold less SE, Alt 5 and 3'SS in the cytoplasmic fraction (~20 fold less for IRs). Are these events also largely retained in the nucleus? This is also related to picking out the IRs in Ln 205 as being 4% in the cytoplasmic. As an example of an alternative group of AS genes, SEs are also very low compared to the nuclear fraction; about 11% cytoplasmic to the nuclear total. All alternative splicing needs to be considered.

Response:

Thank you for your suggestion. IR-induced nuclear localization of the IRTs has been well studied in humans and reported in plants. However, the influences of SE, A5SS, and A3SS on mRNA localization are underexplored. Moreover, in *Arabidopsis*, IR is the most abundant AS event, comprising about 60% of all AS events (Fig. 2). Hence, in this study, we focused on IR events. We have added some discussion on the influences of SE, A5SS, and A3SS in the revised “Results” section.

The authors suggest a role for COP1 regulating gene expression through IR retention in the nucleus and present some evidence showing intron retained transcripts (IRT) are largely retained in the nucleus. Gene expression is under the circadian clock and will show substantial changes in gene expression between light and dark. This will also result in large changes in AS too and will generate numbers of genes showing alternative splicing including IR and the other AS events. The clock is not considered or mentioned here despite the strong evidence that other suppressors of *cop1-6* are clock linked genes.

Response:

Thank you for your suggestion. The circadian clock is the endogenous mechanism that synchronizes the physiological adaptation of an organism to its surroundings based on the day and night transition⁵. It can induce AS of many genes upon light condition changes. Meanwhile, many clock genes are known to undergo alternative splicing in response to changes in light conditions. Taken together, circadian clock and AS form a very complex crosstalk in coordinately regulating plant performance under fluctuating environmental conditions. In this study, we focused on the study of light-/COP1-induced AS in seedling photomorphogenesis. The seedlings in this study were grown under constant dark or constant white light conditions at 22°C. In addition, in Supplemental Fig. S4, GO term related to circadian clock was not enriched. Among the 6 recessive *cop1-6* suppressors, *CSU1–CSU6*, only *CSU4* and *CSU6* are clock-linked genes⁶⁻¹¹. Therefore, we did not mention the circadian clock in the prior manuscript. We have added some discussion on circadian clock in the revised “Results” section.

Ln 209. It is about 50% overlap in the nuclear fraction (Fig3 d) but about 30% in the cytoplasmic fraction (Fig 3e).

Response:

We have revised this typo.

Fig 4 and Ln 213-230 and particularly the summary conclusion line 228-230. I do not see the connection with intron retention. This is not about intron retention and its regulation. This is functional testing of *cop1-6* in other mutant backgrounds. It shows that the *cop1* mutation is the dominant player when it comes to hypocotyl length that

is largely not overcome by overexpression of these other genes. This section needs reviewed.

Response:

This section discusses functional testing of light-/COP1-induced IR regulation. In this study, we thought that IR downregulates gene expression through the detainment of IRTs in the nucleus. WT_L and *cop1-6_d* seedlings, both of which display photomorphogenic phenotypes, had higher SI values for *PIF4*, *RVE1*, and *ABA3* than WT_D (Supplemental Fig. 6c–f and 7d–g). In this case, the proportion of *PIF4*, *RVE1*, and *ABA3* functional transcripts was downregulated in WT_L and *cop1-6_D*, compared with WT_D (Fig. R3). Hence, *pif4-2 rve1-2 aba3-3* displayed a photomorphogenic phenotype with a shorter hypocotyl length than WT_D (Fig. 4a). In addition, *PIF4*, *RVE1*, and *ABA3* greatly increased the hypocotyl length of *cop1-6* in the dark (Fig. 4c). The hypocotyl lengths of *PIF4-YFP myc-RVE1 cop1-6* and *YFP-ABA3 myc-RVE1 cop1-6* were about twice that of *cop1-6*. We speculate that *PIF4-YFP YFP-ABA3 myc-RVE1 cop1-6* may have longer hypocotyl lengths than *PIF4-YFP myc-RVE1 cop1-6* and *YFP-ABA3 myc-RVE1 cop1-6*. However, we did not acquire *PIF4-YFP YFP-ABA3 myc-RVE1 cop1-6* seedlings because both *PIF4-YFP* WT and *YFP-ABA3* WT were bialaphos resistant. We have added some discussion in this section and revised the conclusion sentence.

Ln 233-235. I don't think you can say this. There is still plenty spliced product in your genes and probably many more genes involved. Can you detect these genes in your cytoplasmic and nuclear RNA-seq samples? Splicing occurs on ~80% of plant genes, so alterations of splicing can have a broad effect. Do changes occur in the efficiencies and accuracies of constitutively spliced genes?

Response:

AS is very important to plant growth and development. In *Arabidopsis*, 42–61% of intron-containing genes are subjected to AS^{12, 13, 14}. AS has a broad effect on gene regulation, including affecting the efficiencies and accuracies of constitutively spliced genes. In this study, we found that hundreds of AS events were regulated by both light and COP1. From these genes, we selected four genes for detailed studies. Phenotype analysis in Fig. 4 indicates that, except *PIF4*, *RVE1*, and *ABA3*, many other genes are involved in this process. However, we could not study them all. We also detected that the *PIF4* protein levels in *dcs cop1-6* mutants were higher than in *cop1-6*, which could contribute to the suppressor phenotype of *cop1-6* (Fig. R6). Both *CSU4* and *CSU6* suppressed *cop1-6* by affecting the action of *PIF4*^{9, 11}. The differences in *PIF4* and *RVE1* IRTs were also presented in our nuclear RNA-seq samples.

Fig. R6. PIF4 protein levels in the WT, *cop1-6*, and *dcs cop1-6* mutants. *Arabidopsis* seedlings were grown in the dark for 5 days before harvesting. PIF4 protein levels were checked using the PIF4 antibody (AS16 3955, Agrisera) bought from the company.

Ln 280 and Supp. Fig 7. *dcs1-4* is a T-DNA insertion line, why does supplemental also describe *aba3-3* here as a CRSIPR? Not sure where this fits in and be better to see a model of DCS1-4 T-DNA insertion position.

Response:

Thank you for your suggestion. We have divided Supplemental Fig. 7 into two figures, Supplemental Fig. 8 and Supplemental Fig. 11. We have added the insertion position information for *dcs1-4* in Supplemental Fig. 11.

Ln 286-290. Despite the significant results reported, this can only be described partially at best. Isoginkgetin affects splicing in a general way (preventing stable recruitment of the U4/U5/U6 tri-small nuclear ribonucleoprotein) so it could well be affecting something else that leads to these changes. In addition, the fact that they continue to live means that the level of application of the inhibitor is minimal on splicing. How much drug was used? Not described in methods. Similar argument with Naphthazarin (check spelling – Ln 287).

Response:

Thank you for your suggestion. We agree that the splicing inhibitors partially suppressed the *cop1-6* phenotype. We have revised this typo. We have added the method of splicing inhibitor treatment in the revised “Methods” section. In line 287, ‘naphthazatin’ was changed to ‘naphthazarin’ (line 308 in the revised manuscript).

Ln 303. But the data indicates that it retains all forms of AS gene in the nucleus not just IR.

Response:

Thank you for your suggestion. We have added some discussion about the influences of SE, A5SS, and A3SS in the revised “Results” section.

Ln158-160 and Supp. Fig 4. It is very interesting that GO terms identify genes with a role in pre-mRNA in response to light and these are not present in *cop1-6* affected AS

genes. Further supporting a role for cop1 in splicing and alternative splicing (AS). The authors could add this.

Response:

Thank you for your suggestion. In the revised manuscript, GO terms related to regulation of RNA splicing and regulation of mRNA processing were not enriched in group WT_L vs. WT_D because the mRNA-seq data were reanalyzed using $P < 0.05$ (see revised Supplemental Fig. S4). However, we added some discussion in the revised “Results” section according to your suggestion.

At a technical level.

Ln 164-169 and Fig 2. In relation to Fig2 b,c and f., how much of the IRs reported is noise? A level of IRs may well be partially spliced random products, generally found at low expression values despite being reported as significant differences. Supp Fig 5 validations show splicing indexes for RVE1 and HRD1B as $<5\%$ of the total transcripts. How confident are the authors that the couple of percentage points differences are real responses to light and mutant – despite the significant values reported?

Response:

In addition to the 12 light/COP1-regulated IRTs displayed in the revised manuscript, we also detected 6 IRTs which were not response to both light and *cop1-6* (Fig. R7). Hence, we speculate that around one third of the IR events detected by mRNA-seq are noise. Although SI for *RVE1* and *HRD1B* was low, we checked it with three independent biological replicates. And the differences in SI are also present in the nuclear fraction (Supplemental Fig. 7).

Fig. R7. Intron retention events in validated genes. a Gene structures of *WRKY20*, *WRKY26*, *MYB16*, *SR30*, *IAA7*, and *WEEP*. Red arrows indicate the locations of the primers used to confirm the IR events of each gene through semi-quantitative RT-PCR. The exons are represented by yellow boxes, the UTR regions are

represented by green boxes, and the introns are represented by blue lines. b IR event conformation using semi-quantitative RT-PCR. Red arrows indicate the positions of the IRTs. Blue arrows indicate the positions of the intron-spliced transcripts. The number of PCR cycles used in this analysis were as follows: 25 cycles for *ACT2*, 29 cycles for *WRKY20*, *WRKY26*, *MYB16*, *SR30*, *IAA7*, and *WEEP*.

Ln 409. What sequencing was performed and by who. How were the libraries created. Paired-end sequencing? Result of quality assessment? What are the mapping rates?

Response:

Thank you for your thoughtful review. The libraries were sequenced on the Illumina Novaseq 6000 platform by Novogene Co. Ltd. (Tianjing, China) and 150 bp paired-end reads were generated. The data quality assessment and mapping rates are listed in Table R1 and R2.

Ln 417. The calculation and quantification values of FPKM is not used to compare between sample sets. RNA-seq quantifications are normalised first for gene length, and then sequencing depth with outputs in TPM values that are proportionately comparable between samples.

Response:

Thank you for your kind suggestion. We have made the necessary adjustments to our RNA-Seq data analysis. As you correctly pointed out, we have now standardized the gene expression quantification values to TPM (Transcripts Per Million) after reanalyzing the RNA-Seq data. This normalization process accounts for gene length and sequencing depth differences, thus ensuring that our gene expression values are proportionately comparable between samples.

We want to emphasize that the initial differential expression analysis, which was based on counts, remains unaffected by this change. The list of differentially expressed genes that we previously provided is still valid and reliable for the purposes of our study. Therefore, the results and conclusions drawn from the differential expression analysis remain unchanged.

Supp. Fig 5 a and b. Despite the DNase step in the RNA extraction kit used, how sure are the authors that there is no DNA in the RNA samples used for semi quantitative RT-PCR. I would recommend designing primers across the retained intron and a second spliced intron, that way you will know that the retained intron is not due to contaminating DNA, particularly when the amounts of RT-PCR product is proportionately lower than the spliced products.

Response:

Thank you for your suggestion. We designed primers across the retained intron and a second spliced intron. The results are similar to Supplemental Fig. 6 (Fig. R8).

Fig. R8. Light and COP1 induce changes in the IRTs of *PIF4*, *RVE1*, *ABA3*, and *HRD1B*. a Gene structures of *PIF4*, *RVE1*, *ABA3*, and *HRD1B*. Red arrows indicate the locations of the primers used to confirm the IR events of each gene through semi-quantitative RT-PCR. The exons are represented by yellow boxes, the UTR regions are represented by green boxes, and the introns are represented by blue lines. b IR event conformation using semi-quantitative RT-PCR. Red arrows indicate the positions of IRTs. Blue arrows indicate the positions of intron-spliced transcripts. The number of PCR cycles used in this analysis were as follows: 25 cycles for *ACT2*, 28 cycles for *PIF4*, *RVE1*, and *ABA3*, and 31 cycles for *HRD1B*.

Ln 433-439. Results presented were semi-quantitative PCR reactions run out on a gel. Where are the quantitative real time results? Quantitative data not mentioned in the text other than as a method. How many PCR cycles were used in the semi quantitative RT-PCR analysis. High PCR cycles where the amplification is outwith the exponential phase is not semi-quantitative. It is important to state this.

Response:

Thank you for your suggestion. We have added the number of PCR cycles used in the semi-quantitative RT-PCR analysis in the legends of revised Supplemental Fig. 5, 6, 7, and 11. Moreover, the SI in Supplemental Fig. 6, 7, and 9 was checked using qRT-PCR analysis.

General questions.

It will be interesting to do an RNA-seq experiment that included at least the *cop1-6*, *dcs1* line to determine whether the mutation of DCS1 compensates at least partially for the splicing of the genes affected by the *cop1-6* mutation.

Response:

Thank you for your suggestion. We performed RNA-Seq analysis on dark-grown *cop1-6* and *dcs1-1 cop1-6*. Approximately 66% of AS events in *cop1-6_D* were

recovered in *dcs1-1 cop1-6_D* (Fig. R9).

Fig. R9. Venn diagram illustrating the number of AS events that present in *cop1-6_D* (red), *dcs1-1 cop1-6_D* (green), or in both (yellow). AS, $P < 0.05$.

Do the repressors affect splicing of *cop1-6* that may alters its function?

Response:

The suppressors did not affect the splicing of *COPI-6* mRNA (Fig. R10).

Fig. R10. *dcs* mutations have no effect on the splicing of *COPI-6* mRNA. The *cop1-6* mutation led to four cryptically spliced products at intron 4. PCR products generated from *cop1-6* and *dcs cop1-6* mutants using primers corresponding to the adjacent exons were examined on a 3.5% agarose gel.

The authors say they identified 13 dominant repressors of *cop1-6* and describe 6 of them. What do the other 7 do?

Response:

Through the genetic screen of *cop1-6* suppressors in the dark, we identified and

isolated 13 additional extragenic, dominant suppressors of *cop1-6* (Fig. 1a). Among them, *DCS1* and *DCS3* had three mutant alleles, and *DCS2* had four mutant alleles. Hence, we selected one from each to perform the genomic complementation test and further detailed studies. We have checked the SI of *PIF4*, *RVE1*, *ABA3*, and *HRD1B* in the other 7 *cop1-6* suppressors. The SI of *PIF4*, *RVE1*, *ABA3*, and *HRD1B* were partially recovered in these mutants (Fig. R11).

Fig. R11. The *dcs* suppressors partially rescue the SI of *PIF4*, *RVE1*, *ABA3*, and *HRD1B* in the *cop1-6* mutant. SI of *PIF4* (a), *RVE1* (b), *ABA3* (c), and *HRD1B* (d) in WT_L, WT_D, *cop1-6_D*, and dark-grown *dcs cop1-6* double mutants. * $P < 0.05$, ** $P < 0.01$, *** $P < 0.001$, ns, not significant, as determined by Student's *t*-test.

Minor.

Ln 144. Bit odd to have Fig. 6 before Fig. 2 to 5.

Response:

We apologize for this. We thought that Fig. 6a was not enough to be an independent figure before Figs. 2–5, as it is related to Fig. 6.

References

1. Galej, W.P., Oubridge, C., Newman, A.J. & Nagai, K. Crystal structure of Prp8 reveals active site cavity of the spliceosome. *Nature* **493**, 638-643 (2013).
2. Deng, X. *et al.* Recruitment of the NineTeen Complex to the activated spliceosome requires AtPRMT5. *Proc. Natl Acad. Sci. USA* **113**, 5447-5452 (2016).
3. Chen G.-H., Liu M.-J., Xiong Y., Sheen J. & Wu S.-H. TOR and RPS6 transmit light signals to enhance protein translation in deetioloating *Arabidopsis* seedlings. *Proc. Natl Acad. Sci. USA* **115**, 12823-12828 (2018).
4. Liu M.-J., Wu S.-H., Chen H.-M. & Wu S.-H. Widespread translational control contributes to the regulation of *Arabidopsis* photomorphogenesis. *Mol. Syst. Biol.* **8**, 566-566 (2012).
5. Fan T, *et al.* A crosstalk of circadian clock and alternative splicing under abiotic stresses in the plants. *Front Plant Sci* **13**, 976807 (2022).
6. Xu, D. *et al.* The RING-finger E3 ubiquitin ligase COP1 SUPPRESSOR1 negatively regulates COP1 abundance in maintaining COP1 homeostasis in dark-grown *Arabidopsis* seedlings. *Plant Cell* **26**, 1981-1991 (2014).
7. Xu, D. *et al.* *Arabidopsis* COP1 SUPPRESSOR 2 represses COP1 E3 ubiquitin ligase activity through their coiled-coil domains association. *PLoS Genet.* **11**, e1005747 (2015).
8. Lin, F., Xu, D., Jiang, Y., Chen, H. & Deng, X.W. Phosphorylation and negative regulation of CONSTITUTIVELY PHOTOMORPHOGENIC 1 by PINOID in *Arabidopsis*. *Proc. Natl Acad. Sci. USA* **114**, 6617-6622 (2017).
9. Zhao, X. *et al.* COP1 SUPPRESSOR 4 promotes seedling photomorphogenesis by repressing *CCA1* and *PIF4* expression in *Arabidopsis*. *Proc. Natl Acad. Sci. USA* **115**, 11631-11636 (2018).
10. Zhou, H. *et al.* A missense mutation in WRKY32 converts its function from a positive regulator to a repressor of photomorphogenesis. *New Phytol.* **235**, 111-125 (2022).
11. Lan, H. *et al.* COP1 SUPPRESSOR 6 represses the PIF4 and PIF5 action to promote light-inhibited hypocotyl growth. *J. Integr. Plant Biol.* **64**, 2097-2110 (2022).
12. Filichkin, S.A. *et al.* Genome-wide mapping of alternative splicing in *Arabidopsis thaliana*. *Genome Res.* **20**, 45-58 (2010).
13. Marquez, Y., Brown, J.W.S., Simpson, C., Barta, A. & Kalyna, M. Transcriptome survey reveals increased complexity of the alternative splicing landscape in *Arabidopsis*. *Genome Res.* **22**, 1184-1195 (2012).
14. Wu, H.-P. *et al.* Genome-wide analysis of light-regulated alternative splicing mediated by photoreceptors in *Physcomitrella patens*. *Genome Biol.* **15**, R10-R10 (2014).

Table. R1. Data quality assessment of mRNA-seq data.

sample	raw_reads	raw_bases	clean_reads	clean_bases	error_rate	Q20	Q30	GC_pct
WT_D rep1	41810836	6.27G	40173380	6.03G	0.02	98.14	94.65	45.33
WT_D rep2	47554290	7.13G	45820262	6.87G	0.02	98.1	94.48	45.28
WT_D rep3	43778370	6.57G	42187382	6.33G	0.03	97.89	94.01	45.31
WT_D_C rep1	42604424	6.39G	42190674	6.33G	0.02	98.03	94.29	45.97
WT_D_C rep2	46172802	6.93G	45694466	6.85G	0.02	98.01	94.29	46.21
WT_D_C rep3	39758482	5.96G	38959560	5.84G	0.03	97.47	92.79	46.13
WT_D_N rep1	41698602	6.25G	40057464	6.01G	0.02	98.18	94.66	43.46
WT_D_N rep2	42965062	6.44G	41427960	6.21G	0.02	98.16	94.69	43.4
WT_D_N rep3	42707950	6.41G	41216260	6.18G	0.02	98.3	94.99	43.37
WT_L rep1	41127134	6.17G	39690802	5.95G	0.02	98.28	94.91	46.34
WT_L rep2	47987590	7.2G	45532138	6.83G	0.02	98.35	94.92	46.25
WT_L rep3	44478866	6.67G	42539566	6.38G	0.02	98.32	94.8	46.14
WT_L_C rep1	42341804	6.35G	41448964	6.22G	0.03	97.84	94.11	45.02
WT_L_C rep2	46192496	6.93G	44691916	6.7G	0.03	97.79	93.93	45.75
WT_L_C rep3	46582984	6.99G	45082438	6.76G	0.03	97.54	93.38	43.76
WT_L_N rep1	43122568	6.47G	41111034	6.17G	0.02	98.37	94.85	43.47
WT_L_N rep2	43421860	6.51G	41495326	6.22G	0.02	98.33	94.82	43.38
WT_L_N rep3	41152612	6.17G	40396174	6.06G	0.03	96.98	91.72	43.25
cop1-6_D rep1	41492392	6.22G	39116286	5.87G	0.02	98.46	95.17	45.98
cop1-6_D rep2	41678434	6.25G	38810204	5.82G	0.03	97.89	93.9	46.12
cop1-6_D rep3	41059952	6.16G	38561214	5.78G	0.03	97.84	93.66	45.76
cop1-6_D_C rep1	46677246	7G	44759100	6.71G	0.03	97.5	93.5	46.34
cop1-6_D_C rep2	46413874	6.96G	44802090	6.72G	0.03	97.85	94.09	45.84
cop1-6_D_C rep3	47189576	7.08G	45654412	6.85G	0.03	97.64	93.73	45.75
cop1-6_D_N rep1	41965762	6.29G	40945906	6.14G	0.03	97.43	92.8	43.69
cop1-6_D_N rep2	41965870	6.29G	39564730	5.93G	0.02	98.31	94.83	43.52
cop1-6_D_N rep3	41689726	6.25G	38321122	5.75G	0.03	97.82	93.69	43.45

Table. R2. Mapping rates of mRNA-seq data.

sample	total_reads	total_map	unique_map	multi_map	read1_map	read2_map	positive_map	negative_map	splice_map	unsplice_map	proper_map
WT_D rep1	40173380	39065403(97.24%)	38337231(95.43%)	728172(1.81%)	19172524(47.72%)	19164707(47.7%)	19162072(47.7%)	19175159(47.73%)	15506294(38.6%)	22830937(56.83%)	37382884(93.05%)
WT_D rep2	45820262	44558175(97.25%)	43719530(95.42%)	838645(1.83%)	21900511(47.8%)	21819019(47.62%)	21850586(47.69%)	21868944(47.73%)	17767254(38.78%)	25952276(56.64%)	42636896(93.05%)
WT_D rep3	42187382	40934033(97.03%)	40173681(95.23%)	760352(1.8%)	20146700(47.76%)	20026981(47.47%)	20078594(47.59%)	20095087(47.63%)	16372550(38.81%)	23801131(56.42%)	39085658(92.65%)
WT_D_C rep1	42190674	40024281(94.87%)	37784901(89.56%)	2239380(5.31%)	18923313(44.85%)	18861588(44.71%)	18890226(44.77%)	18894675(44.78%)	14349245(34.01%)	23435656(55.55%)	37265642(88.33%)
WT_D_C rep2	45694466	42173285(92.29%)	39683779(86.85%)	2489506(5.45%)	19887545(43.52%)	19796234(43.32%)	19839623(43.42%)	19844156(43.43%)	15117710(33.08%)	24566069(53.76%)	39116898(85.61%)
WT_D_C rep3	38959560	36774387(94.39%)	34127363(87.6%)	2647024(6.79%)	17113412(43.93%)	17013951(43.67%)	17059277(43.79%)	17068086(43.81%)	12590311(32.32%)	21537052(55.28%)	33583542(86.2%)
WT_D_N rep1	40057464	38928112(97.18%)	37336133(93.21%)	1591979(3.97%)	18698719(46.68%)	18637414(46.53%)	18658868(46.58%)	18677265(46.63%)	12664539(31.62%)	24671594(61.59%)	36363148(90.78%)
WT_D_N rep2	41427960	40168044(96.96%)	38569031(93.1%)	1599013(3.86%)	19322297(46.64%)	19246734(46.46%)	19275985(46.53%)	19293046(46.57%)	13132756(31.7%)	25436275(61.4%)	37556992(90.66%)
WT_D_N rep3	41216260	40058406(97.19%)	38576538(93.6%)	1481868(3.6%)	19321544(46.86%)	19264384(46.74%)	19279354(46.78%)	19297184(46.82%)	13289692(32.24%)	25286846(61.35%)	37597028(91.22%)
WT_L rep1	39690802	38776663(97.7%)	38038469(95.84%)	738194(1.86%)	19053576(48.01%)	18984893(47.83%)	19013346(47.9%)	19025123(47.93%)	15485784(39.02%)	22552685(56.82%)	37131850(93.55%)
WT_L rep2	45532138	44433930(97.59%)	43523613(95.59%)	910317(2.0%)	21778270(47.83%)	21745343(47.76%)	21754531(47.78%)	21769082(47.81%)	17698668(38.87%)	25824945(56.72%)	42319332(92.94%)
WT_L rep3	42539566	41567220(97.71%)	40786830(95.88%)	780390(1.83%)	20408650(47.98%)	20378180(47.9%)	20387852(47.93%)	20398978(47.95%)	16808380(39.51%)	23978450(56.37%)	39658426(93.23%)
WT_L_C rep1	41448964	38837730(93.7%)	35487989(85.62%)	3349741(8.08%)	17818583(42.99%)	17669406(42.63%)	17731438(42.78%)	17756551(42.84%)	10510450(25.36%)	24977539(60.26%)	34754618(83.85%)
WT_L_C rep2	44691916	39632967(88.68%)	39110595(87.51%)	522372(1.17%)	19639708(43.94%)	19470887(43.57%)	19541848(43.73%)	19568747(43.79%)	13352273(29.88%)	25758322(57.64%)	38309434(85.72%)
WT_L_C rep3	45082438	40215314(89.2%)	39708019(88.08%)	507295(1.13%)	20013004(44.39%)	19695015(43.69%)	19836496(44.0%)	19871523(44.08%)	11884308(26.36%)	27823711(61.72%)	38697696(85.84%)
WT_L_N rep1	41111034	40244508(97.89%)	39060574(95.01%)	1183934(2.88%)	19542190(47.54%)	19518384(47.48%)	19520582(47.48%)	19539992(47.53%)	13637607(33.17%)	25422967(61.84%)	38058406(92.57%)
WT_L_N rep2	41495326	40614131(97.88%)	39347316(94.82%)	1266815(3.05%)	19679860(47.43%)	19667456(47.4%)	19664376(47.39%)	19682940(47.43%)	14187279(34.19%)	25160037(60.63%)	38450568(92.66%)
WT_L_N rep3	40396174	39429985(97.61%)	38246019(94.68%)	1183966(2.93%)	19234438(47.61%)	19011581(47.06%)	19103122(47.29%)	19142897(47.39%)	12335926(30.54%)	25910093(64.14%)	37404064(92.59%)
cop1-6_D rep1	39116286	38288419(97.88%)	36964102(94.5%)	1324317(3.39%)	18482271(47.25%)	18481831(47.25%)	18476199(47.23%)	18487903(47.26%)	15134862(38.69%)	21829240(55.81%)	35999092(92.03%)
cop1-6_D rep2	38810204	37641394(96.99%)	36785303(94.78%)	856091(2.21%)	18440876(47.52%)	18344427(47.27%)	18378314(47.35%)	18406989(47.43%)	15086946(38.87%)	21698357(55.91%)	35535110(91.56%)
cop1-6_D rep3	38561214	37571273(97.43%)	36776201(95.37%)	795072(2.06%)	18470637(47.9%)	18305564(47.47%)	18378714(47.66%)	18397487(47.71%)	15197637(39.41%)	21578564(55.96%)	35633850(92.41%)
cop1-6_D_C rep1	44759100	35791719(79.97%)	35320073(78.91%)	471646(1.05%)	17836742(39.85%)	17483331(39.06%)	17646050(39.42%)	17674023(39.49%)	12688140(28.35%)	22631933(50.56%)	34382984(76.82%)
cop1-6_D_C rep2	44802090	34707779(77.47%)	34267864(76.49%)	439915(0.98%)	17223635(38.44%)	17044229(38.04%)	17116762(38.21%)	17151102(38.28%)	12077666(26.96%)	22190198(49.53%)	33526994(74.83%)
cop1-6_D_C rep3	45654412	37889552(82.99%)	37440348(82.01%)	449204(0.98%)	18859304(41.31%)	18581044(40.7%)	18705577(40.97%)	18734771(41.04%)	12892822(28.24%)	24547526(53.77%)	36534204(80.02%)
cop1-6_D_N rep1	40945906	39596097(96.7%)	37464770(91.5%)	2131327(5.21%)	18794714(45.9%)	18670056(45.6%)	18713845(45.7%)	18750925(45.79%)	13478262(32.92%)	23986508(58.58%)	36157612(88.31%)
cop1-6_D_N rep2	39564730	38680021(97.76%)	36679547(92.71%)	2000474(5.06%)	18356474(46.4%)	18323073(46.31%)	18333928(46.34%)	18345619(46.37%)	12725549(32.16%)	23953998(60.54%)	35637790(90.07%)
cop1-6_D_N rep3	38321122	37048716(96.68%)	35367636(92.29%)	1681080(4.39%)	17734853(46.28%)	17632783(46.01%)	17667481(46.1%)	17700155(46.19%)	12677684(33.08%)	22689952(59.21%)	33933550(88.55%)

REVIEWER COMMENTS

Reviewer #1 (Remarks to the Author):

This reviewer appreciates the authors' diligent efforts in correcting and improving the methods, data presentation, and discussion in the revised manuscript. There are two suggestions left to be addressed before this manuscript is considered for acceptance.

1. Results in Fig. R1 in the rebuttal letter are helpful for discussing the possible mechanisms for these mutant alleles to suppress *cop1-6* in the aspect of spliceosome/proteasome function. Including the results in Fig. R1 (if they are reproducible) can better strengthen the mechanistic insights.

2. In the response letter (but not in the revised manuscript), the authors described "The distribution of the mutation sites in DCS proteins is within highly conserved regions among different species." From Fig. 1c, mutations for *dcs2-3*, *2-4* were not in highly conserved domains? Multiple sequence alignment should be provided to state they are within conserved regions among different species. It's also under whether the authors have examined their presentation on protein topology as for *dcs3* mutants (Supplemental Fig. 10). If no structural data is available, perhaps at least examine AlphaPulldown?

Reviewer #2 (Remarks to the Author):

I reviewed this paper previously and thank the authors for responding and answering the majority of my questions and showing experimental data in support. I also recognise the work and effort that has gone into this paper. It is very intriguing that around half of the suppressors found in this *cop1-6* mutational screen are linked to the spliceosome, which suggest a splicing role. Some of these genes are key (eg *PRP8*) to the function of the spliceosome and splicing in general. The authors consideration of splicing/alternative splicing as a response to suppression of the *cop1* mutant is therefore logical.

Splicing occurs in the majority of plant genes and so it can be expected to have wide effects on a broad range of genes. Different introns have variation in efficiencies and regulatory elements that can alter splice site selection and the efficient removal of an intron leading to regulation of individual gene function. Intron retention is one of three other main AS events that regulates the

functional expression of a gene. The data here shows general splicing remains efficient and accurate as can be seen in the multi-intron gene examples presented (eg Figure 1 f) but a few thousand genes show significant differential alternative splicing. The differences in differential gene expression is not presented here.

I still have difficulties with the main hypothesis that suppressors of the cop1-6 dark grown photomorphogenic response and the light signal is a result of intron retentions based on the data presented. It may well have a role, but there is a need to consider all aspects of splicing that includes the other alternative events and what genes result in the return to etiolation in the dark, as shown in the repressors. Just over half of the AS events identified are AS events are intron retentions which means just under half are alternative events, which is still a significant number (Fig 2 D). The authors response says that about a third of these intron retention events may be partially spliced introns, which suggests that more than half of the other three AS events presented are responding to the change. The argument in the responses that says less is known about the other events is true in relation to what is known about intron retention transcripts retained in the nucleus. It does not mean that these other AS are not important and they are relatively easy to identify and characterise. To highlight this response as intron retention important only does not consider all the possible AS events and the genes they are found in. I still think the authors need to consider all the AS events and what genes they are found in.

The authors have selected a number of genes in the main text, in the supplementary and in the response. I understand the gene selection used for this part of the study and see the differences in intron retention, but these genes show good levels of transcript that would result in functional protein translation. It is hard in my view to see how the intron retained would contribute to the suppression of translation of a functional protein, generated from these transcripts. I take on board the argument presented in the response, Figure R3 that functional transcript is reduced to a level that indicates statistical significance, but it is hard to say this level of reduction is the cause of the suppression. In supplemental Figure 7b the functional transcript levels of the selected genes look very similar in the cytoplasmic data, which makes it difficult to indicate an intron retention response to light in the Wt light and cop1-6 dark. Based on Supplemental Figure 7b there is also an apparent transcriptional difference, which may aid the observed decrease in splicing index in the Wt dark sample shown in Supplemental Figure 7 d-g. So what are the significantly altered genes and AS genes that change the cop1-6 mutant to a light type growth type in the dark? These differences need to be considered more than just the observation that these gene have increased/decreased retention. The other genes described in supplemental Fig 5 showing largely similar intron retention levels between cop1-6 grown in the dark and wt plants grown in the light. This is shown as an increase (eg BBX17) or decrease (eg. CKG) in intron retention. But the level of transcription varies between genes and many of the genes still have good levels of properly spliced transcript.

The transcriptome analysis includes a carefully prepared nuclear v cytoplasmic RNA-seq datasets which gives a great opportunity to study the gene transcripts that are available for translation. I understand that intron retention is the more abundant AS event described in plants and that many/most are retained in the nucleus. But the RNA-seq data between nuclear and cytoplasmic shows large differences in numbers and would indicate many transcripts are in the nucleus. What do other As events do. Does just one AS transcript make it out to the cytoplasm or does the alternative event also make it out?

In summary, the work done on this is excellent, the datasets are valuable, but I think the interpretation is only partial. I think the authors have not found the complete reason for the suppression yet and I don't think a regulation of intron retained transcripts between light and dark will be the only cause of the photomorphogenic phenotype. The authors have the RNAseq data to do a complete transcriptome analysis that will help identify the important genes and splicing regulated genes.

Response letter to reviewers' comments

REVIEWER COMMENTS

Reviewer #1 (Remarks to the Author):

This reviewer appreciates the authors' diligent efforts in correcting and improving the methods, data presentation, and discussion in the revised manuscript. There are two suggestions left to be addressed before this manuscript is considered for acceptance.

1. Results in Fig. R1 in the rebuttal letter are helpful for discussing the possible mechanisms for these mutant alleles to suppress cop1-6 in the aspect of spliceosome/proteasome function. Including the results in Fig. R1 (if they are reproducible) can better strengthen the mechanistic insights.

Response:

Thank you for the suggestion. We have added these data in the revised "Discussion" section; and yes, those observations are rather consistent.

2. In the response letter (but not in the revised manuscript), the authors described "The distribution of the mutation sites in DCS proteins is within highly conserved regions among different species." From Fig. 1c, mutations for *dcs2-3*, 2-4 were not in highly conserved domains? Multiple sequence alignment should be provided to state they are within conserved regions among different species.

Response:

Thank you for the suggestion. As suggested, we did the multiple sequence alignment of DCS2. The result shows that the point mutations in *dcs2-1*, *dcs2-2*, *dcs2-3*, and *dcs2-4* are within the conserved regions (Fig. NR1).

Figure NR1. Sequence alignment of the conserved DCS2 regions containing G367E, A457S, G495R, and D512N mutations in *dcs2-1*, *dcs2-2*, *dcs2-3*, and *dcs2-4*, respectively. The point mutation sites in *dcs2* mutants are labeled by red triangles.

It's also unclear whether the authors have examined their presentation on protein topology as for *dcs3* mutants (Supplemental Fig. 10). If no structural data is available, perhaps at least examine AlphaPulldown?

Response:

DCS3, the counterpart of yeast Prp8, is highly conserved across different species except for the N-terminus¹. Based on the recently reported atomic structure of yeast Prp8 (885-2413 aa), the mutated amino acids of the *dcs3-2* and *dcs3-3* alleles are on the outer surface of the RT/En domain, a crucial protein-binding surface, which may affect the interactions of DCS3 with surrounding components within the spliceosome (Supplemental Fig. 10)^{1, 2}. Currently, it is a little difficult for us to analyze the differences in the protein topologies between DCS3 and DCS3-1, DCS3-2, DCS3-3 proteins using AlphaPulldown. Instead, we predicted the protein structures of DCS3, DCS3-1, DCS3-2, and DCS3-3 using SWISS-MODEL, a fully automated protein structure homology-modelling server³. The results show that the mutations in *dcs3-2* and *dcs3-3*, but not *dcs3-1*, bring changes to the protein structure of DCS3 (Fig. NR2, Supplemental Fig. 13 in the revised manuscript). These structural changes might affect the interactions of DCS3 with surrounding components within the spliceosome.

Fig. NR2. The point mutations in *dcs3-2* and *dcs3-3* change the protein structure of DCS3. a-d The protein structures of DCS3 (a), DCS3-1 (b), DCS3-2 (c), and DCS3-3 (d) predicted by SWISS-MODEL using the reported human Prp8 structure (SMTL ID: 5ygz.1) as template. e PyMOL alignment of DCS3, DCS3-1, DCS3-2, and DCS3-3 protein structures. The protein structure changes induced by the point mutations in *dcs3-2* and *dcs3-3* are marked in red box.

Reviewer #2 (Remarks to the Author):

I reviewed this paper previously and thank the authors for responding and answering the majority of my questions and showing experimental data in support. I also recognise the work and effort that has gone into this paper. It is very intriguing that around half of the suppressors found in this cop1-6 mutational screen are linked to the spliceosome, which suggest a splicing role. Some of these genes are key (eg PRP8) to the function of the spliceosome and splicing in general. The authors consideration of splicing/alternative splicing as a response to suppression of the cop1 mutant is therefore logical.

Splicing occurs in the majority of plant genes and so it can be expected to have wide effects on a broad range of genes. Different introns have variation in efficiencies and regulatory elements that can alter splice site selection and the efficient removal of an intron leading to regulation of individual gene function. Intron retainment is one of three other main AS events that regulates the functional expression of a gene. The data here shows general splicing remains efficient and accurate as can be seen in the multi-intron gene examples presented (eg Figure 1 f) but a few thousand genes show significant differential alternative splicing. The differences in differential gene expression is not presented here.

Response:

Thank you for the suggestion. Alternative splicing is a crucial mechanism for gene expression regulation through generating multiple mature mRNA isoforms. Among the four main AS events, IR is the most common AS events in plants. IR regulates gene expression through nuclear detention of the intron-retained transcripts (IRTs)^{4, 5, 6}. We have now included the expression data for the representative genes: *PIF4*, *RVE1*, *ABA3*, and *HRD1B* (Fig. NR3).

Fig. NR3. Gene expression of *PIF4*, *RVE1*, *ABA3*, and *HRD1B*. Total RNA was extracted from 5-d-old WT_L, WT_D, and *cop1-6_D* seedlings. Quantitative real-time PCR was used to check the expression of *PIF4* (a), *RVE1* (b), *ABA3* (c), and *HRD1B* (d). * $P < 0.05$, ** $P < 0.01$, *** $P < 0.001$, ns, not significant, as determined by Student's t-test.

I still have difficulties with the main hypothesis that suppressors of the *cop1-6* dark grown photomorphogenic response and the light signal is a result of intron retentions based on the data presented. It may well have a role, but there is a need to consider all aspects of splicing that includes the other alternative events and what genes result in the return to etiolation in the dark, as shown in the repressors. Just over half of the AS events identified are AS events are intron retentions which means just under half are alternative events, which is still a significant number (Fig 2 D). The authors response says that about a third of these intron retention events may be partially spliced introns, which suggests that more than half of the other three AS events presented are responding to the change.

Response:

In our analysis, IR is the most prevalent event, representing around 60% of total AS events (Fig. 2b, c). We paid great attention to the role of IR in the regulation of seedling photomorphogenic development, while the other AS events were also carefully examined. Previous studies have reported that IR leads to nuclear detention of the IRTs, and then avoid degradation by nonsense-mediated mRNA decay (NMD)^{4,5,6}. However, the ES, A3SS, and A5SS transcripts can be transported to the cytoplasm, where many of them are degraded by NMD⁴. Hence, IR events function in a different way from the ES, A3SS, and A5SS events in this regard. And this difference prompts us to focus on IR events in this study. Nevertheless, we agree that the ES, A3SS, and A5SS events may also play a role in the regulation of seedling photomorphogenic development, but in different mechanisms.

To confirm the mRNA-seq data, we selected 18 IR events that were present in both WT_L vs. WT_D and *cop1-6_D* vs. WT_D groups for experimental validation using semiquantitative RT-PCR analysis. We found that 6 of the 18 IR events cannot be

repeated in the semiquantitative RT-PCR analysis. Hence, it is speculated that around one third of the IR events detected by mRNA-seq are noise. Based on this, we also speculate that around one third of the other three AS events detected by mRNA-seq are noise.

The argument in the responses that says less is known about the other events is true in relation to what is known about intron retention transcripts retained in the nucleus. It does not mean that these other AS are not important and they are relatively easy to identify and characterise. To highlight this response as intron retention important only does not consider all the possible AS events and the genes they are found in. I still think the authors need to consider all the AS events and what genes they are found in.

Response:

Thanks for your suggestion. In our analysis, we analyzed all the four kinds of AS events. GO enrichment analysis showed that GO term related to ‘response to light stimulus’ was significantly enriched in IR genes, but not significantly enriched in A3SS, A5SS, and ES genes (Fig. NR4, NR5). However, GO terms related to ‘RNA splicing’ and ‘RNA processing’ were significantly enriched in A3SS, A5SS, and ES genes, especially in genes derived from WT_L vs. WT_D group (Fig. NR4, NR5). Hence, we proposed that the regulation of IR genes is more likely to contribute to the phenotype differences of young *Arabidopsis* seedlings between light and darkness.

In addition, we focused on the IR events, because IR works in a different way. IR leads to nuclear detention of the IRTs, and then avoid degradation by NMD^{4, 5, 6}. However, the ES, A3SS, and A5SS transcripts can be transported to the cytoplasm, in where many of them are degraded by NMD⁴. Upon external stimulus and developmental phase changes, the unspliced introns in IRTs can be removed post-transcriptionally in a spliceosome-dependent and transcription-independent manner^{5, 7}. After that, the fully spliced transcript isoforms translocate into the cytoplasm and are competent for translation^{5, 7}. Although our data supported a role of IR in light control of seedling development, we are not excluding potential roles of other AS events, in addition to the major role of COP1 regulation of transcription factors abundance directly.

Fig. NR4. GO analysis of IR (top 40 GO terms) (a), A3SS (top 40 GO terms) (b), A5SS (c), and ES genes (d) identified by mRNA-seq in WT_L vs. WT_D group.

Fig. NR5. GO analysis of IR (top 40 GO terms) (a), A3SS (b), A5SS (c), and ES (d) genes identified by mRNA-seq in *cop1-6_D* vs. WT_D group.

The authors have selected a number of genes in the main text, in the supplementary and in the response. I understand the gene selection used for this part of the study and see the differences in intron retention, but these genes show good levels of transcript that would result in functional protein translation. It is hard in my view to see how the intron

retained would contribute to the suppression of translation of a functional protein, generated from these transcripts. I take on board the argument presented in the response, Figure R3 that functional transcript is reduced to a level that indicates statistical significance, but it is hard to say this level of reduction is the cause of the suppression.

Response:

Alternative splicing is a crucial mechanism for gene expression regulation through generating multiple mature mRNA isoforms. It is supposed that AS genes can also have good levels of properly spliced transcripts. For example, Shikata et al., found that light, depending on phytochrome, induce the enhancement of splice variants of *SPA3* (from about 55% to 60%) to promote photomorphogenesis⁸. These splice variants of *SPA3* were thought to encode truncated *SPA3* proteins that have dominant-negative effects on the function of the endogenous COP1–SPA complex⁸. And in this case, *SPA3* also have good levels of properly spliced transcript⁸. In our study, though mRNA-seq, we identified that the IRTs of *PIF4*, *RVE1*, *ABA3*, and *HRD1B* are in response to light and *cop1-6*. The splicing index (SI) values for *PIF4*, *RVE1*, and *ABA3* were higher in WT_L and *cop1-6_D* seedlings than in WT_D seedlings, while the SI value for *HRD1B* was higher in WT_D seedlings (Supplemental Fig. 6). And these give rise to significant changes in the functional transcripts of *PIF4*, *RVE1*, *ABA3*, and *HRD1B* among WT_L, *cop1-6_D* and WT_D seedlings (Fig. NR6, Supplemental Fig. 7 in the revised manuscript), which are supposed to contribute to the phenotype changes among WT_L, *cop1-6_D* and WT_D. We agree that IR changes in any single gene may not be sufficient for the phenotype changes, but accumulative effects of IRs from many genes could bring about observable phenotype changes.

In Supplemental Fig. 7b, in the cytoplasmic fraction, the expression of *PIF4* and *RVE1* is higher in WT_D than in WT_L and *cop1-6_D* seedlings. We also carried on quantitative real-time PCR to confirm it. The results show that, in the cytoplasmic fraction, the expression of *PIF4*, *RVE1*, and *ABA3* is higher in WT_D than in WT_L and *cop1-6_D* seedlings (Fig. NR7, Supplemental Fig. 9 in the revised manuscript). In addition, we detected that the PIF4 protein levels in dark-grown WT and *dcs cop1-6* mutants were higher than in dark-grown *cop1-6*, which could contribute to the suppressor phenotype of *cop1-6* (Fig. NR8).

Fig. NR6. Light and COP1 induce changes in the functional transcripts of *PIF4* (a), *RVE1* (b), *ABA3* (c), and *HRD1B* (d). Total RNA was extracted from 5-d-old WT_L, WT_D, and *cop1-6_D* seedlings. The exons are represented by yellow boxes, the UTR regions are represented by green boxes, and the introns are represented by blue lines. The red boxes represent the intron-retaining exons. mRNA-1 represents the splice variant considered to encode the functional full-length protein, while mRNA-2 represents the intron-containing splice variant identified by mRNA-seq. The values are shown as the mean \pm SE (n = 3). Asterisks indicate statistical significance, as determined using Student's *t*-test (* P < 0.05; ** P < 0.01).

Fig. NR7. The expression of *PIF4* (a), *RVE1* (b), *ABA3* (c), and *HRD1B* (d) in the cytoplasmic fraction of WT_L, WT_D, and *cop1-6_D* seedlings. Cytoplasmic RNA was extracted from 5-d-old WT_L, WT_D, and *cop1-6_D* seedlings. The values are shown as the mean \pm SE (n = 3). Asterisks indicate statistical significance, as determined using Student's *t*-test (* P < 0.05; ** P < 0.01, *** P < 0.001, ns, not significant).

Fig. NR8. PIF4 protein levels in the WT, *cop1-6*, and *dcs cop1-6* mutants. *Arabidopsis* seedlings were grown in the dark for 5 days before harvesting. PIF4 protein levels were checked using the PIF4 antibody (AS16 3955, Agrisera) bought from the company.

In supplemental Figure 7b the functional transcript levels of the selected genes look very similar in the cytoplasmic data, which makes it difficult to indicate an intron retention response to light in the WT light and *cop1-6* dark. Based on Supplemental Figure 7b there is also an apparent transcriptional difference, which may aid the observed decrease in splicing index in the WT dark sample shown in Supplemental Figure 7 d-g. So what are the significantly altered genes and AS genes that change the *cop1-6* mutant to a light type growth type in the dark? These differences need to be considered more than just the observation that these genes have increased/decreased retention.

Response:

In Supplemental Figure 7b, we observed the transcriptional differences. However, the SI is defined as the abundance of the intron-retained isoform relative to the total mRNA level. Hence, only the transcriptional differences should not cause the SI differences. The SI values for *PIF4*, *RVE1*, and *ABA3* were higher in WT_L and *cop1-6*_D seedlings than in WT_D seedlings, while the SI value for *HRD1B* was higher in WT_D seedlings (Supplemental Fig. 6). And these give rise to significant changes in the functional transcripts of *PIF4*, *RVE1*, *ABA3*, and *HRD1B* among WT_L, *cop1-6*_D and WT_D seedlings (Fig. NR6).

The other genes described in supplemental Fig 5 showing largely similar intron retention levels between *cop1-6* grown in the dark and wt plants grown in the light. This is shown as an increase (eg BBX17) or decrease (eg. CKG) in intron retention. But the level of transcription varies between genes and many of the genes still have good levels of properly spliced transcript.

Response:

The genes listed in Fig. S5 are used to confirm the mRNA-seq data. All the IRTs of these genes are in response to light and *cop1-6*. We think these AS genes can have good levels of proper spliced transcripts. In our case, thousands of genes have AS events. If these AS genes cannot have good levels of proper spliced transcripts, the growth and development of *Arabidopsis* seedlings will be severely affected.

The transcriptome analysis includes a carefully prepared nuclear v cytoplasmic RNA-seq datasets which gives a great opportunity to study the gene transcripts that are available for translation. I understand that intron retention is the more abundant AS event described in plants and that many/most are retained in the nucleus. But the RNA-seq data between nuclear and cytoplasmic shows large differences in numbers and would indicate many transcripts are in the nucleus. What do other AS events do. Does just one AS transcript make it out to the cytoplasm or does the alternative event also make it out?

Response:

AS coupled to NMD is an important way in regulation of gene expression⁴. The AS transcripts can be targeted for degradation by NMD in the cytoplasm⁴. Although IR is the most common AS event in plants, the intron-retained transcripts are not sensitive to NMD due to nuclear detention^{4, 5, 6}. While those ES, A3SS, and A5SS transcripts can be transported into the cytoplasm, in where many of them are degraded by NMD⁴.

In summary, the work done on this is excellent, the datasets are valuable, but I think the interpretation is only partial. I think the authors have not found the complete reason for the suppression yet and I don't think a regulation of intron retained transcripts between light and dark will be the only cause of the photomorphogenic phenotype.

Response:

Thank you for the suggestion. We agree that the regulation of intron-retained transcripts between light and dark is not the only cause of the photomorphogenic phenotype. In addition to give rise to the genome-wide alternative splicing changes, light also induces nuclear reorganization, chromatin remodeling, global reprogramming in transcriptome and translome, and so on^{8, 9, 10, 11, 12, 13, 14}. All these changes contribute to the phenotype differences of young *Arabidopsis* seedlings between light and dark. Hence, there is no doubt that the regulation of intron-retained transcripts between light and dark is not the only cause of the photomorphogenic phenotype. In this study, we identified that the IRTs of *PIF4*, *RVE1*, *ABA3*, and *HRD1B* are in response to light and *cop1-6*. And the changes in the SI values for *PIF4*, *RVE1*, *ABA3*, and *HRD1B* give rise to statistically significant changes in the functional transcripts of *PIF4*, *RVE1*, *ABA3*, and *HRD1B* among WT_L, *cop1-6_D* and WT_D seedlings (Supplemental Fig. 6 and Fig. NR6), which are suggested to contribute to the phenotype changes between light and dark (Fig. 4). This mechanism has not been reported before.

The authors have the RNAseq data to do a complete transcriptome analysis that will help identify the important genes and splicing regulated genes.

Response:

Thank you for the suggestion. We have done a complete transcriptome analysis of our RNAseq data. In our analysis, although we cannot exclude that changes of some other genes contribute to the phenotype differences of young *Arabidopsis* seedlings between light and dark, we think the regulation of *PIF4*, *RVE1*, and *ABA3* plays an important role in this process. We found that the IRTs of *PIF4*, *RVE1*, *ABA3*, and *HRD1B* are in response to light and *cop1-6*. And the changes in the SI values for *PIF4*, *RVE1*, *ABA3*, and *HRD1B* give rise to significant changes in the functional transcripts of *PIF4*, *RVE1*, *ABA3*, and *HRD1B* among WT_L, *cop1-6*_D and WT_D seedlings (Supplemental Fig. 6 and Fig. NR6). Further, we found that *PIF4*, *RVE1*, and *ABA3* are regulators of photomorphogenesis, and they contribute to the phenotype differences of young *Arabidopsis* seedlings between light and dark (Fig. 4). Hence, we selected them for detailed studies.

In a previous study, Huang et al., also found that light can induce the changes in IR variants of *HY5*, *PIF3*, *PIF4*, *BBX22*, and *BBX24* during dark to light treatment¹⁵. In their study, they focused only on IR events and selected *BBX22* and *BBX24* for detailed studies. However, the IR events of *HY5*, *PIF3*, *BBX22*, and *BBX24* were not isolated in our study, probably due to different treatment methods. Moreover, we hope we can discover some other genes or mechanisms involved in seedling photomorphogenic development in our future work.

References

1. Galej WP, Oubridge C, Newman AJ, Nagai K. Crystal structure of Prp8 reveals active site cavity of the spliceosome. *Nature* **493**, 638-643 (2013).
2. Deng X, *et al.* Recruitment of the NineTeen Complex to the activated spliceosome requires AtPRMT5. *Proc. Natl Acad. Sci. USA* **113**, 5447-5452 (2016).
3. Biasini M, *et al.* SWISS-MODEL: modelling protein tertiary and quaternary structure using evolutionary information. *Nucleic Acids Res.* **42**, W252-258 (2014).
4. Kalyna M, *et al.* Alternative splicing and nonsense-mediated decay modulate expression of important regulatory genes in *Arabidopsis*. *Nucleic Acids Res.* **40**, 2454-2469 (2012).
5. Jia J, *et al.* Post-transcriptional splicing of nascent RNA contributes to widespread intron retention in plants. *Nat. Plants* **6**, 780-788 (2020).
6. Göhring J, Jacak J, Barta A. Imaging of endogenous messenger RNA splice variants in living cells reveals nuclear retention of transcripts inaccessible to nonsense-mediated decay in *Arabidopsis*. *Plant Cell* **26**, 754-764 (2014).
7. Boothby Thomas C, Zipper Richard S, van der Weele Corine M, Wolniak Stephen M. Removal of retained introns regulates translation in the rapidly developing gametophyte of *Marsilea vestita*. *Dev. Cell* **24**, 517-529 (2013).
8. Shikata H, Hanada K, Ushijima T, Nakashima M, Suzuki Y, Matsushita T. Phytochrome controls alternative splicing to mediate light responses in *Arabidopsis*. *Proc. Natl Acad. Sci. USA* **111**, 18781-18786 (2014).
9. Bourbousse C, *et al.* Light signaling controls nuclear architecture reorganization during seedling establishment. *Proc. Natl Acad. Sci. USA* **112**, E2836-E2844 (2015).
10. Feng CM, Qiu Y, Van Buskirk EK, Yang EJ, Chen M. Light-regulated gene repositioning in *Arabidopsis*. *Nat. Commun.* **5**, 3027 (2014).
11. Charron J-BF, He H, Elling AA, Deng XW. Dynamic landscapes of four histone modifications during deetiolation in *Arabidopsis*. *Plant cell* **21**, 3732-3748 (2009).
12. Ma L, *et al.* Genomic evidence for COP1 as a repressor of light-regulated gene expression and development in *Arabidopsis*. *Plant Cell* **14**, 2383-2398 (2002).
13. Liu M-J, Wu S-H, Chen H-M, Wu S-H. Widespread translational control contributes to the regulation of *Arabidopsis* photomorphogenesis. *Mol. Syst. Biol.* **8**, 566-566 (2012).
14. Jang G-J, Yang J-Y, Hsieh H-L, Wu S-H. Processing bodies control the selective translation for optimal development of *Arabidopsis* young seedlings. *Proc. Natl Acad. Sci. USA* **116**, 6451-6456 (2019).
15. Huang C-K, Lin W-D, Wu S-H. An improved repertoire of splicing variants and their potential roles in *Arabidopsis* photomorphogenic development. *Genome Biol.* **23**, 50 (2022).

REVIEWER COMMENTS

Reviewer #2 (Remarks to the Author):

I have read through the paper again and think it still fundamentally flawed by the poor consideration of other regulated AS events.

I do not think this a reason not to publish, as the focus on IRs is important and relevant but not to consider the other events in my view diminishes the completeness of the paper and the hard work done by the authors.

Considering the responses to my previous questions.

1. The presentation of the total gene expression helps to show that, despite relatively constant gene expression, the proportion of unspliced to spliced has changed. This supports the splicing change hypothesis given by the authors.

2. Consideration of all aspects of splicing. The authors quite rightly distinguish IRs as showing nuclear retention, while the other AS transcripts by exon skipping, alternative 5' splice site and alternative 3' splice site may be expected to be transported and many transcripts may well be targeted for NMD and degraded and therefore not seen. But, this of course means that any of these AS transcripts that are significantly changing and not degraded may well be important. If they are significantly differentially spliced and present they are showing this despite NMD. Whether the spliced transcripts make it out of the nucleus or not is irrelevant, the splicing process occurs in the nucleus and if this is affecting splicing then it will affect potentially all forms of AS transcripts the ones retained in the nucleus or the ones that make it to translation without NMD degradation and observable. This point applies to another nuclear v cytoplasmic response that uses NMD as an argument.

The GO analysis. This is a light related experiment, so it is not surprising light related genes are found. Splicing factor genes are well known to show alternative splicing and, in some cases, regulate their own transcripts. So it is not surprising this is in a 'Top 40' GO grouping for these.

My argument therefore remains unchanged, other events should be considered.

3. Expression levels. The PIF4 western analysis is convincing showing the suppression of the cop1-6 mutant. The control HSP82 also looks like it is higher in the suppressor background and how this relates to the transcriptional levels is quite difficult comparing Fig NR8 with Fig. NR6 /Fig. Supplemental 7, without considering protein turnover.

Dominant negatives are not an argument when considering nuclear retained IR transcripts.

4. Spliced index (SI). The authors response was ‘..the SI is defined as the abundance of the intron-retained isoform relative to the total mRNA level. Hence, only the transcriptional differences should not cause the SI differences. ‘ This works only if the amount of total transcript produced by all the transcripts stays the same. This is usually not the case and AS occurs in constantly shifting transcription and splicing levels. The case I was describing, did show some transcriptional differences that would affect SI levels. I wanted the authors to respond saying they considered significant SI in the context of all the transcripts. Do they?

Response letter to reviewers' comments

REVIEWER COMMENTS

Reviewer #2 (Remarks to the Author):

1. The presentation of the total gene expression helps to show that, despite relatively constant gene expression, the proportion of unspliced to spliced has changed. This supports the splicing change hypothesis given by the authors.

Response:

Thanks for your comments.

2. Consideration of all aspects of splicing. The authors quite rightly distinguish IRs as showing nuclear retention, while the other AS transcripts by exon skipping, alternative 5' splice site and alternative 3' splice site may be expected to be transported and many transcripts may well be targeted for NMD and degraded and therefore not seen. But, this of course means that any of these AS transcripts that are significantly changing and not degraded may well be important. If they are significantly differentially spliced and present they are showing this despite NMD. Whether the spliced transcripts make it out of the nucleus or not is irrelevant, the splicing process occurs in the nucleus and if this is affecting splicing then it will affect potentially all forms of AS transcripts the ones retained in the nucleus or the ones that make it to translation without NMD degradation and observable. This point applies to another nuclear v cytoplasmic response that uses NMD as an argument.

The GO analysis. This is a light related experiment, so it is not surprising light related genes are found. Splicing factor genes are well known to show alternative splicing and, in some cases, regulate their own transcripts. So it is not surprising this is in a 'Top 40' GO grouping for these.

My argument therefore remains unchanged, other events should be considered.

Response:

Thanks for your comments. We completely agree that the ES, A3SS, and A5SS events might also play an important role in the regulation of seedling photomorphogenic development. As suggested, we have carefully analyzed these three AS events. We found that GO term related to 'response to light stimulus' was not enriched in ES, A3SS, or A5SS genes, but significantly enriched in IR genes (Fig. R1, R2). That's one reason why we focus on IR events in this study. More importantly, we focused on IR events, because it functions in a different way as the other three AS events. Previous studies have reported that IR leads to nuclear detention of the IRTs, and then avoid degradation by nonsense-mediated mRNA decay (NMD)^{1, 2, 3}. Upon external stimulus and developmental phase changes, the unspliced introns in IRTs can be removed post-transcriptionally in a spliceosome-dependent and transcription-independent manner, and thus rapidly responding to changes in environmental signals^{2, 4}. After that, the fully

spliced transcript isoforms translocate into the cytoplasm and are competent for translation⁴. However, the ES, A3SS, and A5SS transcripts can be transported to the cytoplasm, where many of them are degraded by NMD¹. A recent study published in *PNAS* also focused on the function of the post-transcriptionally spliced introns in seedling photomorphogenic development and confirmed that COP1 plays an important role in this process⁵.

Fig. R1. GO analysis of IR (top 40 GO terms) (a), A3SS (top 40 GO terms) (b), A5SS

(c), and ES genes (d) identified by mRNA-seq in WT_L vs. WT_D group.

Fig. R2. GO analysis of IR (top 40 GO terms) (a), A3SS (b), A5SS (c), and ES (d) genes identified by mRNA-seq in *cop1-6_D* vs. WT_D group.

3. Expression levels. The PIF4 western analysis is convincing showing the suppression of the *cop1-6* mutant. The control HSP82 also looks like it is higher in the suppressor

background and how this relates to the transcriptional levels is quite difficult comparing Fig NR8 with Fig. NR6 /Fig. Supplemental 7, without considering protein turnover.

Dominant negatives are not an argument when considering nuclear retained IR transcripts.

Response:

Thanks for your suggestion. In this study, though mRNA-seq, we identified that the IRTs of *PIF4*, *RVE1*, *ABA3*, and *HRD1B* are in response to light and *cop1-6*. The splicing index (SI) values for *PIF4*, *RVE1*, and *ABA3* were higher in WT_L and *cop1-6_D* seedlings than in WT_D seedlings, while the SI value for *HRD1B* was higher in WT_D seedlings (Supplemental Fig. 6). And these give rise to significant changes in the functional transcripts of *PIF4*, *RVE1*, *ABA3*, and *HRD1B* among WT_L, *cop1-6_D* and WT_D seedlings (Supplemental Fig. 7), and thus inducing protein level changes. And in Fig. NR8 (see also in Fig. R3a), the PIF4 protein levels is significantly higher in dark-grown WT and *dcs cop1-6* mutants than in dark-grown *cop1-6*, which could contribute to the suppressor phenotype of *cop1-6*. For considering protein turnover, we treated WT, *cop1-6*, and *dcs cop1-6* seedlings with protease inhibitors MG132 and E64d. After MG132 and E64d treatments, we also detected that the PIF4 protein levels in dark-grown WT and *dcs cop1-6* mutants were higher than in dark-grown *cop1-6*, indicating that these changes might be caused by the SI changes (Fig. R3b, c).

Fig. R3. PIF4 protein levels in the WT, *cop1-6*, and *dcs cop1-6* mutants. a *Arabidopsis* seedlings were grown in the dark for 5 days before harvesting. b and c *Arabidopsis*

seedlings were grown in the dark for 5 days before protease inhibitor treatment. Seedlings were treated with 50 μ M MG132 (b) or 20 μ M E64d (c) for 3 h in the dark, respectively.

4. Spliced index (SI). The authors response was ‘..the SI is defined as the abundance of the intron-retained isoform relative to the total mRNA level. Hence, only the transcriptional differences should not cause the SI differences. ‘ This works only if the amount of total transcript produced by all the transcripts stays the same. This is usually not the case and AS occurs in constantly shifting transcription and splicing levels. The case I was describing, did show some transcriptional differences that would affect SI levels. I wanted the authors to respond saying they considered significant SI in the context of all the transcripts. Do they?

Response:

Thanks for your suggestion. We have compared the light- and COP-regulated IR genes with the light- and COP-regulated differentially expressed genes. We found that only a small proportion of the light- and COP-regulated IR genes overlapped with the light- and COP-regulated differentially expressed genes, indicating that the most light- and COP-regulated IR events were not caused by the differential expression (Fig. R4). In addition, to consider significant SI in the context of all the transcripts, we did transcript quantification analysis of all the transcripts using the RNA-seq data as shown in Table. R1. In this analysis, we also found that the IR transcripts of *PIF4*, *ABA3*, and *HRD1B* show significant differences among WT_L, WT_D, and *cop1-6_D*. However, we did not find the differences in the IR transcripts of *RVE1*, which displayed significant differences in the AS analysis using rMATS (v4.1.2) and the qPCR analysis (Supplemental Fig. 6 and Supplemental Data 1).

Fig. R4. Venn diagram showing the number of genes that displayed both light- and COP1-dependent changes in IR, DE, or in both IR and DE.

References

1. Kalyna M, *et al.* Alternative splicing and nonsense-mediated decay modulate expression of important regulatory genes in *Arabidopsis*. *Nucleic Acids Res.* **40**, 2454-2469 (2012).
2. Jia J, *et al.* Post-transcriptional splicing of nascent RNA contributes to widespread intron retention in plants. *Nat. Plants* **6**, 780-788 (2020).
3. Göhring J, Jacak J, Barta A. Imaging of endogenous messenger RNA splice variants in living cells reveals nuclear retention of transcripts inaccessible to nonsense-mediated decay in *Arabidopsis*. *Plant Cell* **26**, 754-764 (2014).
4. Boothby Thomas C, Zipper Richard S, van der Weele Corine M, Wolniak Stephen M. Removal of retained introns regulates translation in the rapidly developing gametophyte of *Marsilea vestita*. *Dev. Cell* **24**, 517-529 (2013).
5. Yan Y, *et al.* Light controls mesophyll-specific post-transcriptional splicing of photoregulatory genes by AtPRMT5. *Proc. Natl Acad. Sci. USA* **121**, e2317408121 (2024).